# PCR-like performance of rapid test with permselective tunable nanotrap

Seong Jun Park[1,12], Seungmin Lee[1,2,12], Dongtak Lee[2,12], Na Eun Lee[1,3], Jeong Soo Park[1], Ji Hye Hong[1,2], Jae Won Jang[2,4], Hyunji Kim[2,4], Seokbeom Roh[5,6], Gyudo Lee[5,6], Dongho Lee[7], Sung-Yeon Cho[8,9], Chulmin Park[8], Dong-Gun Lee[8,9], Raeseok Lee[8,9], Dukhee Nho[8,9], Dae Sung Yoon[2,4,10,13] ✉, Yong Kyoung Yoo[11,13] ✉ & Jeong Hoon Lee[1,13] ✉

Highly sensitive rapid testing for COVID-19 is essential for minimizing virus transmission, especially before the onset of symptoms and in asymptomatic cases. Here, we report bioengineered enrichment tools for lateral flow assays (LFAs) with enhanced sensitivity and specificity (BEETLES²), achieving enrichment of SARS-CoV-2 viruses, nucleocapsid (N) proteins and immunoglobulin G (IgG) with 3-minute operation. The limit of detection is improved up to 20-fold. We apply this method to clinical samples, including 83% with either intermediate (35%) or low viral loads (48%), collected from 62 individuals ($n = 42$ for positive and $n = 20$ for healthy controls). We observe diagnostic sensitivity, specificity, and accuracy of 88.1%, 100%, and 91.9%, respectively, compared with commercial LFAs alone achieving 14.29%, 100%, and 41.94%, respectively. BEETLES², with permselectivity and tunability, can enrich the SARS-CoV-2 virus, N proteins, and IgG in the nasopharyngeal/oropharyngeal swab, saliva, and blood serum, enabling reliable and sensitive point-of-care testing, facilitating fast early diagnosis.

Fast screening and testing for COVID-19 enables the identification of infected high-risk individuals and, consequently, reduces virus spread, providing better prevention and control of COVID-19[1,2]. Rapid detection and treatment with proper quarantine are the best strategies for handling pandemics. For commercial rapid testing of COVID-19, 59 antigen diagnostic tests for SARS-CoV-2 are available under Emergency Use Authorization (EUA) (as of 1 Jan 2023)[3]. Notwithstanding the developed techniques with advanced diagnostic performance[4–13],

reverse transcription-quantitative polymerase chain reaction (RT–qPCR) tests are still considered the gold standard for COVID-19 diagnosis. Although a highly sensitive assay can be realized using RNA detection technologies, on-site frequent tests of COVID-19 are challenging to perform. RT-qPCR has high cost, long operation time (4–6 h), and long turnaround times (up to several days)[14–17]. Moreover, the highly sensitive and selective RT-qPCR technique requires expensive laboratory equipment and cumbersome RNA extraction steps,

[1]Department of Electrical Engineering, Kwangwoon University, 20 Kwangwoon-ro, Nowon, Seoul 01897, Republic of Korea. [2]School of Biomedical Engineering, Korea University, 145 Anam-ro, Seongbuk, Seoul 02841, Republic of Korea. [3]Department of Biotechnology, College of Life Sciences and Biotechnology, Korea University, Seoul 02841, Republic of Korea. [4]Interdisciplinary Program in Precision Public Health, Korea University, Seoul 02841, Republic of Korea. [5]Department of Biotechnology and Bioinformatics, Korea University, Sejong 30019, Republic of Korea. [6]Interdisciplinary Graduate Program for Artificial Intelligence Smart Convergence Technology, Korea University, Sejong 30019, Korea. [7]CALTH Inc., Changeop-ro 54, Seongnam, Gyeonggi 13449, Republic of Korea. [8]Vaccine Bio Research Institute, College of Medicine, The Catholic University of Korea, Seoul, Republic of Korea. [9]Division of Infectious Diseases, Department of Internal Medicine, College of Medicine, The Catholic University of Korea, Seoul, Republic of Korea. [10]Astrion Inc, Seoul 02841, Republic of Korea. [11]Department of Electronic Engineering, Catholic Kwandong University, 24, Beomil-ro 579 beon-gil, Gangneung-si, Gangwon-do 25601, Republic of Korea. [12]These authors contributed equally: Seong Jun Park, Seungmin Lee, Dongtak Lee. [13]These authors jointly supervised this work: Dae Sung Yoon, Yong Kyoung Yoo, Jeong Hoon Lee. ✉e-mail: dsyoon@korea.ac.kr; yongkyoung0108@cku.ac.kr; jhlee@kw.ac.kr

limiting its application in point-of-care tests (POCTs)[18]. Long turn-around times allow the infected people to spread the virus exponentially before getting results, limiting the impact of isolation and contact tracing[8,19]. Importantly, asymptomatic people spread the virus to communities with a similar viral load to those who develop symptoms[20].

To handle COVID-19 spread, the best approach is fast, easy, on-site detection at the beginning of infection to stop onward spread with high-frequency tests. In contrast to routinely tested RT-qPCR, the high-frequency test should be low-cost and point-of-care. Mina et al. has claimed that the best COVID-19 filter can be achieved using frequent, low-cost, simple, and rapid tests because SARS-CoV-2 quickly grows and spreads out exponentially[19]. A one-time monitored highly sensitive RT-qPCR test can detect viral shedding long after the infectious period (about 9 days), up to 17 days[21], i.e., the convalescence stage with less transmission.

Regarding high-frequency tests for decentralized local testing, the lateral flow assay (LFA) platform is confirmed as the best platform for point-of-care and self-testing as they provide convenient, easy-to-use, one-step, disposable, and fast results[22]. The LFA platforms are considered the best candidate as they meet the WHO (World Health Organization) "ASSURED" criteria (affordable, sensitive, specific, user-friendly, rapid and robust, equipment-free, and deliverable to end-users)[23]. The lower LFA performance limits the accuracy, increases the false negatives, and hampers the early diagnosis of COVID-19. Generally, the LFA platform shows good accuracy for high viral loads; however, accuracy decreases abruptly with low viral loads. In addition, immunoassay-based LFA tests are less sensitive than molecular-based RT-qPCR, generating more false negatives and increasing the risk of onward transmission of the virus. Recently, Chen et al. reported a quantitative and ultrasensitive in situ immunoassay using a nanoporous anodic aluminum oxide (AAO) membrane, showing the ability to enrich SARS-CoV-2 viruses[9]. Generally, AAO membranes have been considered a promising material to facilitate size-based separation and enrichment[24–26].

Here, we develop bioengineered enrichment tools for LFA with enhanced sensitivity and specificity (BEETLES[2]) combined with a commercial COVID-19 LFA kit. A key component of BEETLES[2] is a permselective tunable nanotrap that can enhance commercial LFA's clinical performance. We prepare BEETLES[2] with a combination of red blood cell membranes (RBCMs) and nanoporous anodic aluminum oxide (AAO). The combination of high AAO flux and an aquaporin (AQP) water channel, unlike size-based AAO membrane enrichment[27,28], allows fast water transport with additional permselectivity within 3 min, achieving enrichment of SARS-CoV-2 viruses, nucleocapsid (N) proteins, and immunoglobulin G (IgG) antibodies. The limit of detection (LOD) is enhanced up to 20-fold. Moreover, we show enhanced diagnostic sensitivity, specificity, and accuracy when applied to patient samples.

## Results and discussion
### Nanotrap for enhancing COVID-19 diagnostics (BEETLES[2])
We demonstrate the key concept of BEETLES[2], which enriches and separates biomolecules to sizes and charges, and its application for immuno and molecular assays (Fig. 1). A hybrid filter consisting of RBCM on an AAO membrane is called the BEETLES[2] membrane. In the following sections, we describe more details about the permselective tunable enrichment of BEETLES[2] with various criteria such as size, charge, and pressure. Figure 1a depicts a prototype for POCT sample preparation (details in Fig. S1). Equipped with the BEETLES[2] membrane, we enriched the target samples and enhanced the assays for POCT and molecular diagnostics. For COVID-19 applications, we focused on the enrichment for N proteins, SARS-CoV-2 viruses, and IgG antibodies. The N protein, one of the four structural proteins of the SARS-CoV-2 virus, is the target protein for most COVID-19 Ag LFAs. The IgG

antibodies are the target molecules for COVID-19 Ab LFA. Figure 1b shows the assay protocol from sampling to the assay. After sample collection, we enriched the target molecule via BEETLES[2], applied a well-established diagnostic tool such as commercial LFA and RT-qPCR, and enhanced assay performance (Supplementary Video). Since BEETLES[2] can enrich N proteins, SARS-CoV-2 viruses, and IgG antibodies, BEETLES[2]-assisted LFA enables the daily monitoring of infectious diseases.

The primary advantage of BEETLES[2] is that the viral protein and virus itself can be enriched under various buffers, including phosphate-buffered saline (PBS), saliva, serum, and viral transport medium (VTM). Moreover, abundant protein (albumin) and small inhibitors can be eliminated. Larger biomolecules, like IgG (~150 kDa), are enriched because larger proteins cannot freely pass through the BEETLES[2] membrane. Interestingly, we observed the permselectivity of BEETLES[2] membrane with the N proteins (~46 kDa) and bovine serum albumin (BSA) (~66.5 kDa) proteins. The N protein was positively charged under physiological conditions, whereas BSA was negatively charged. Therefore, we only enriched the N proteins while the negative charged BSA freely passed through the BEETLES[2] membrane and was filtered out.

### Physicochemical characterization of the nanotrap
Figure 2 presents the physical and chemical characterization of the BEETLES[2] membrane. It consists of RBCM and AAO, which are key hybrid materials for enrichment. A honeycomb-like nanostructured AAO membrane with an aligned 20 nm pore was used as a substrate (Fig. 2a). The AAO membrane itself has no permselectivity and is responsible for size-based separation. To add tunable permselectivity, we functionalized RBCM onto the AAO membrane (Fig. 2b). Since the RBCM acts as a physical barrier, we observe no tunable permselectivity in the BEETLES[2] membrane without pressure. However, we observed tunable permselectivity with an applied pressure >0.5 bar.

We functionalized the AAO membrane with RBCM via the vesicle fusion method (Fig. 2c, Figs. S2 and 3). The conformational characteristics of the BEETLES[2] membrane were characterized using scanning electron microscopy (SEM), atomic force microscopy (AFM), and fluorescent microscopy. The uncoated AAO membrane showed a mesh-like network, which aided water and protein penetration. Conversely, the BEETLES[2] membrane had a smooth surface, indicating that RBCM suspension completely covered the porous structure and formed a multilayered RBCM structure (~1 μm on the AAO membrane, side view).

From the SEM images (Fig. S4), we observed conformal coating of RBCM with the least defect over 2% RBCM concentration. 4% RBCM also showed conformal coating; however, it slowed water transport compared to 2% RBCM. For additional topological analysis, we measured the surface roughness of the BEETLES[2] membrane depending on the RBCM concentration (0 – 4% (v/v)) via AFM (Fig. 2d, S5 and 6). Over the scanned $10 \times 10$ μm area, the surface root means square roughness (Rq) of each BEETLES[2] membrane significantly decreased compared to that of a bare AAO membrane, indicating more conformal RBCM coating on the surface of AAO membrane.

Along with AFM and SEM analysis, we conducted Kelvin probe force microscopy (KPFM) (Fig. 2e, f and S7). Via KPFM, we measured the physicochemical characterizations of the BEETLES[2] membrane, which was negatively charged due to the negatively charged phospholipids in RBCM[29,30]. In contrast, the BEETLES[2] membrane with 4% RBCM showed a relatively positively charged surface compared to that with 2% RBCM. The flow of current between the KPFM cantilever tip and sample is speculated to have slowed down because the non-conductive RBCM was excessively wrapped in a dried state on the surface of the AAO membrane[31]. We selected 2% RBCM as the optimal condition because a negatively charged surface potential of the

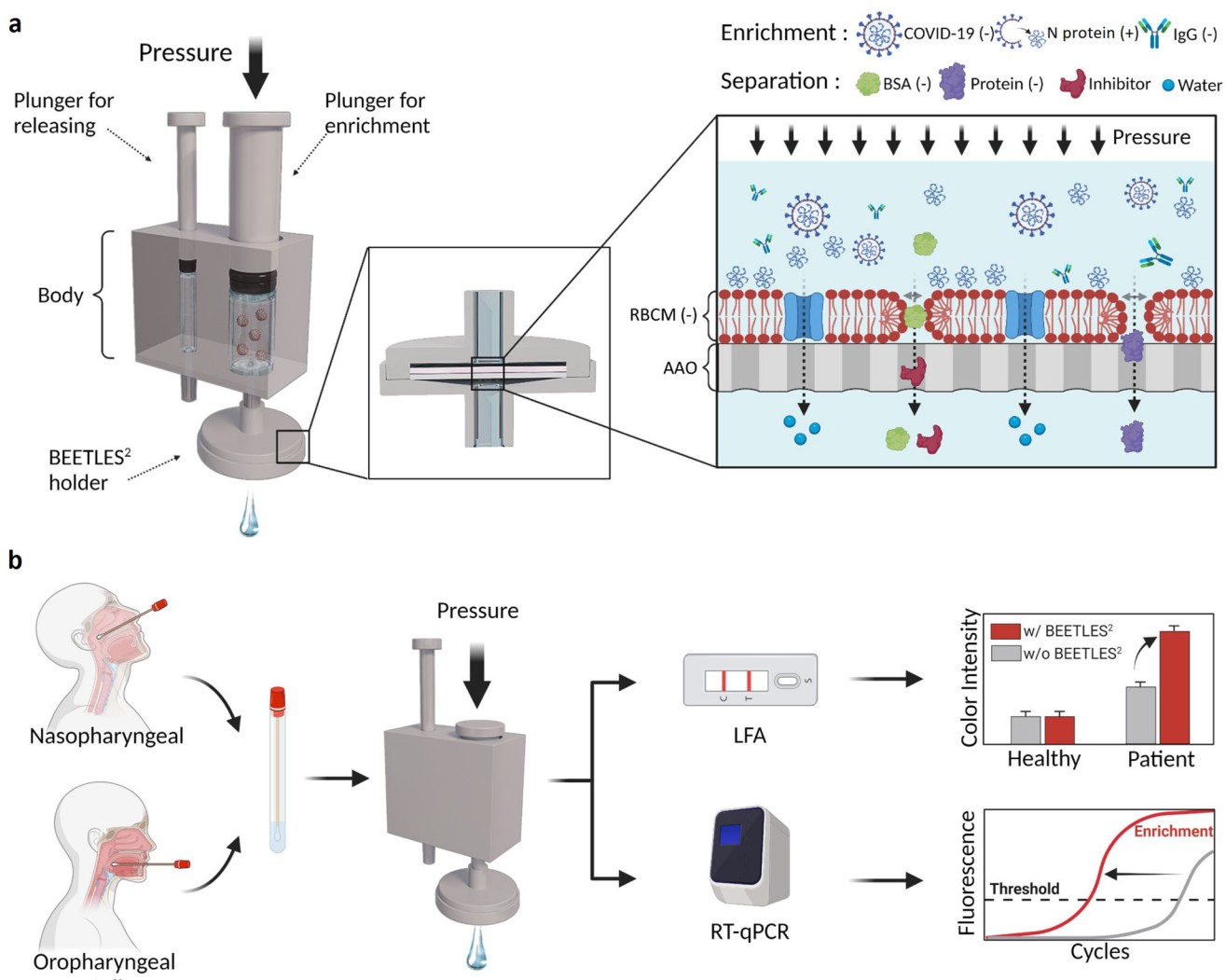

**Fig. 1 | Illustration of BEETLES² for COVID-19 diagnostics. a** BEETLES², a hybrid filter consisting of RBCM on an AAO membrane, has permselectivity and tunability that enable biomolecules to enrich with a separation function. **b** Assay with BEE-TLES² for immunoassay and molecular assay. Cartoons in panels **a, b** were created with BioRender.com. BEETLES², bioengineered enrichment tools for the LFA with enhanced sensitivity and specificity. AAO anodic aluminum oxide, LFA lateral flow assay, RT-qPCR reverse transcription quantitative polymerase chain reaction, RBCM red blood cell membrane.

BEETLES² membrane is significant for maximizing the electrostatic interaction with the positively charged nucleocapsid protein.

To cross-check the optimal RBCM concentration, we analyzed the fluorescent images with different RBCM concentrations (0–4% (v/v)). Fluorescent lipids (PE-CF) were intercalated into RBCM vesicles and coated on AAO membranes. Next, fluorescent images were measured (Fig. S8). Based on topological analysis, KPFM, and fluorescent data, 2% RBCM concentration was determined to be the best for the BEETLES² membrane. Additionally, we validated fluorescence recovery after photobleaching (FRAP) and measured the lateral diffusivity and mobility of RBCM on the AAO membrane (Fig. 2g). Fluorescence recovery across the photobleached spots (spot radius: 20 μm) at the focal plane confirmed the formation of mobile and contiguous lipid bilayers on the AAO membrane. We calculated the mobile fraction (MF) and 2D diffusion coefficient (D) from fluorescence intensity recovery over time. The bleached fluorescence gradually restored up to approximately 80% of the original value within 150 s of laser removal. We determined that the cell membrane translational diffusion coefficient was 0.83 μm² s⁻¹, consistent with lipids in intact RBCs (D = 0.82 μm² s⁻¹)[32,33]. This result indicates that RBCM preserves its lateral diffusivity and fluidity in 2D planar deposition.

## Tunability and permselectivity of the BEETLES² membrane

Figure 3 validates the performance of the BEETLES² membrane in achieving fast permselective enrichment with the aid of fast water transport (flux: 221.61 Lm⁻² h⁻¹). Additionally, the AQP water-transporting proteins on RBCM provide a key route for permselective enrichment (Fig. 3a)[34]. We classified BEETLES² membrane enrichment depending on pressure, molecular size, charge, and buffer conditions. The AAO membrane has a large pore size that allows various proteins such as albumin, viral protein, and immunoglobulin to pass through. However, the BEETLES² membrane can selectively block these proteins because RBCM acts as a physical barrier. Interestingly, with external pressure, negatively charged and small molecules (e.g., albumin) can easily penetrate the BEETLES² membrane owing to the high mobility of RBCM[32,35].

First, to elucidate the pressure effect on enrichment in terms of tunability, we operated BEETLES² under various pressures (Fig. 3b–f). The error bar in Fig. 3 represents the run-to-run deviation (n = 3). We used NanoDrop and sodium dodecyl sulfate-polyacrylamide gel electrophoresis (SDS-PAGE) to measure enrichment. Negatively charged BSA and positively charged N proteins were tested under a physiological condition. With positively charged N proteins, enrichment

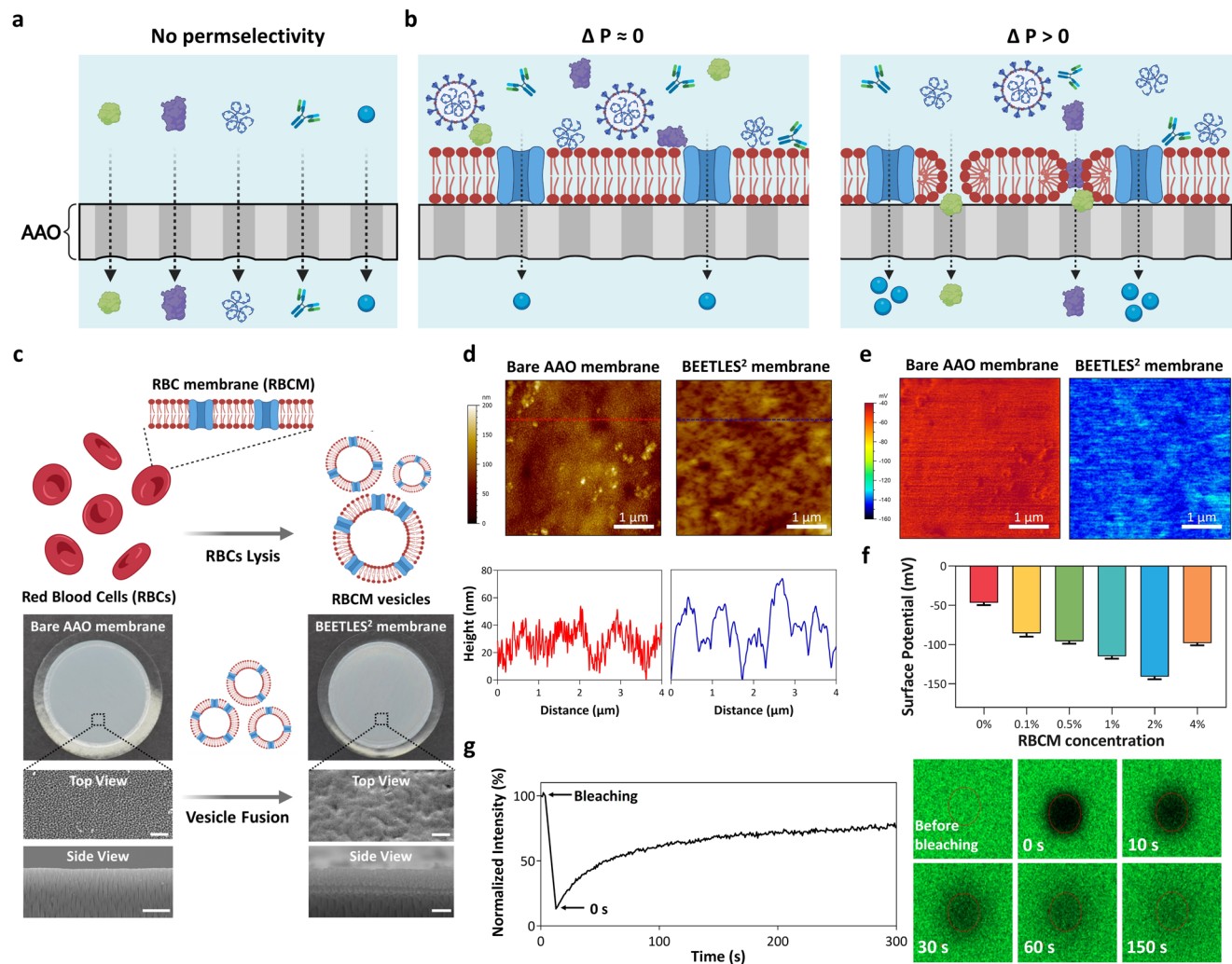

**Fig. 2 | Physicochemical characterization of BEETLES² membrane. a** AAO membrane with an aligned 20 nm pore as a substrate, showing no permselectivity. **b** Functionalization of RBCM onto the AAO membrane, showing permselectivity and tunability under applied pressure. **c** Fabrication and topological analysis of BEETLES² membrane. **d** AFM images showing topography and cross-sectional profiles from the height map images of bare AAO and BEETLES² membrane. (**e, f**) KPFM analysis of (**e**) surface potential images of bare AAO and BEETLES² membrane and **f** Frequencies of surface potential depending on RBCM concentration (0-4% (v/v)). Data are from three pooled experiments (0% RBCM, *n* = 30; 0.1-4% RBCM, *n* = 50). **g** FRAP study of fluorescence intensity with time and images showing RBCM on the substrate is well preserved its lateral diffusivity and fluidity in 2D planar deposition and the recovery of ~80% of the original value was done in 150 s (spot radius: 20 μm). Error bars represent standard deviation from the mean. Cartoons in panels **a**–**c** were created with BioRender.com. BEETLES², bioengineered enrichment tools for the LFA with enhanced sensitivity and specificity; AAO nanoporous anodic aluminum oxide, LFA lateral flow assay, RBCM red blood cell membrane, AFM atomic force microscopy, FRAP fluorescence recovery after photobleaching, KPFM Kelvin probe force microscopy.

exceeded 24-fold measured by nanodrop regardless of pressure (0.1–3 bar). In contrast, negatively charged BSA showed strong dependency on pressure (Fig. 3b). Especially for BSA samples with lower pressure (0.1 bar), a significant increase in enrichment was observed due to no permeation of BSA. The most probable explanation is cell squeezing that enables delivery of a diversity of materials[36], representing the need for applying pressure for negatively charged protein permeation. With increasing pressure, BSA freely passed through the BEETLES²' membrane, without significant enrichment. Importantly, with tunable properties under pressure, we efficiently removed the most abundant protein, i.e., albumin, and small inhibitors. From SDS-PAGE, BSA was enriched under 0.1 bar while passing through over 1 bar, clearly confirming enrichment tunability under different pressures (Fig. 3c).

Second, we demonstrated the enrichment of IgG, with a molecular weight (MW) of 150 kDa, the most common antibody in blood and other body fluids (Fig. 3d). The daily monitoring of IgG could be a crucial test for convalescence in COVID-19. Owing to size filtration, we enriched IgG up to 18-fold regardless of the applied pressure (0.1–3 bar). To validate the size effect of IgG, we tested Human IgG Fc fragments (MW: 50 kDa) and observed no enrichment of IgG Fc fragments at a pressure of 3 bar, inferring the size effect of negatively charged IgG on the enrichments (Fig. S9). The pressure was controlled using regulator-controlled nitrogen gas. We enriched 3 mL of each sample into a final volume of 100 μL, theoretically corresponding to 30-fold enrichment. IgG enrichment was confirmed under a pressure of 3 bar (SDS-PAGE in Fig. 3e). Unlike an antigen or virus-based assay, for serological rapid diagnostic tests, only 1-2 drops (<100 μL) of blood was used. To handle tens of μL of samples, a new design is needed. One such prospective design could be the BEETLES² membrane-integrated microfluidic system.

Third, we checked the surface charge effect on the permselectivity and enrichment of biomolecules (Fig. 3f). Under physiological conditions, N proteins (MW: 46 kDa) with isoelectric points (pI) of 10.3–10.7 exhibited an overall positive charge, whereas BSA (MW: 66.5 kDa) had a pI of 4.5–4.8, leading to a negative charge. The applied

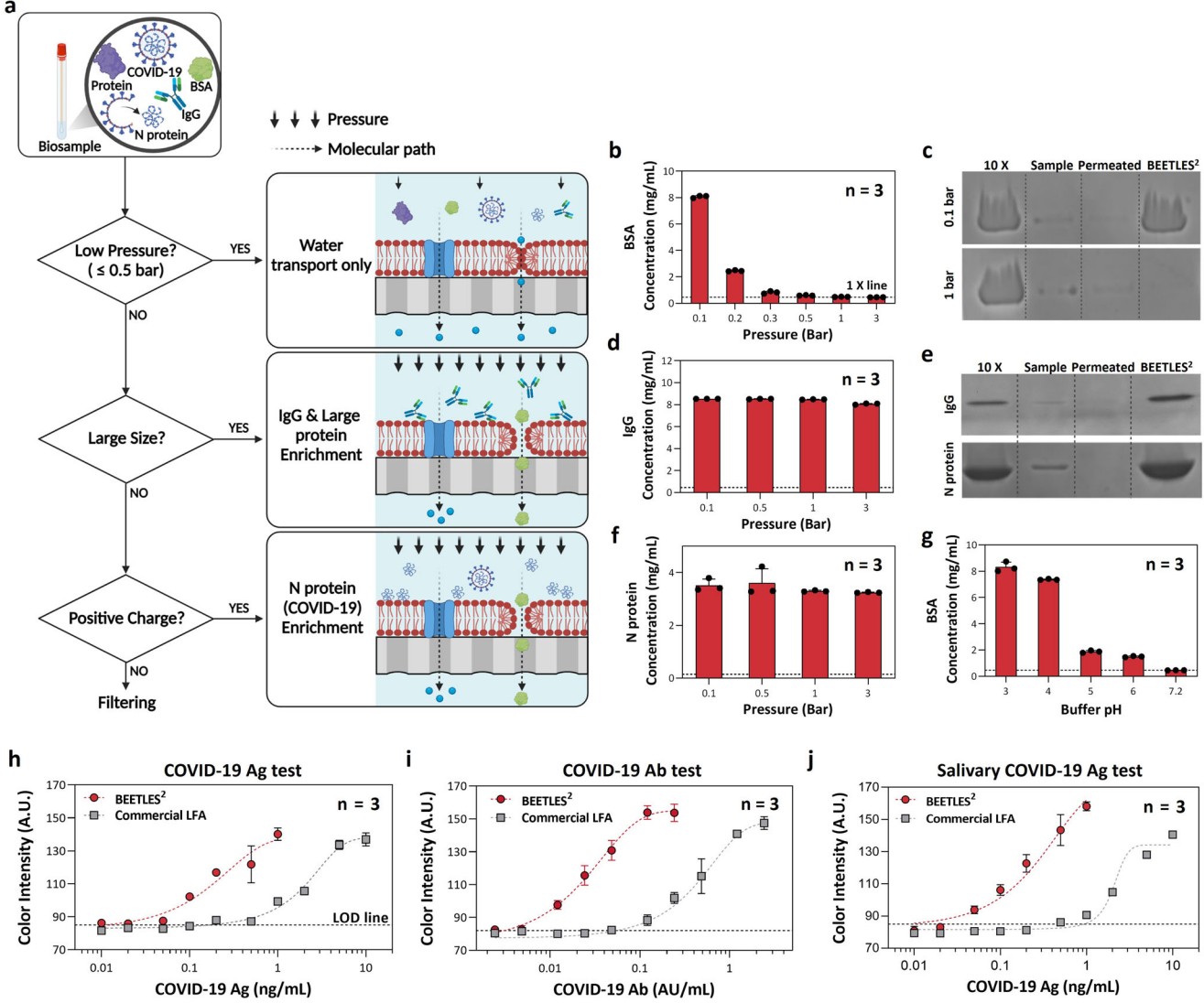

**Fig. 3 | Assay performance validation. a** Flowchart showing the permselectivity and tunability of BEETLES[2] membrane by pressure, molecular size, charge, and buffer conditions. **b** Pressure effect on the enrichment: negatively charged BSA with a strong dependency on pressure, showing tunable properties under pressure. **c** SDS-PAGE of BSA (MW: 66.5 kDa) **d** Size-based enrichment: Immunoglobulin G (IgG). **e** SDS-PAGE of IgG (MW: 150 kDa) and N Protein (MW: 46 kDa). **f** Surface charge-based enrichment (permselectivity): The enrichment with positively charged N proteins. **g** Surface charge effect (permselectivity): The enrichment with positively charged BSA (pI: 4.5-4.8) at lower pH, while no enrichment at higher pH (neutral and negatively charged conditions). (**h, j**) Enhanced sensitivity via BEETLES[2] with (**h**) COVID-19 Ag test, (**i**) COVID-19 IgG rapid test, and (**j**) salivary COVID-19 Ag test. Error bars represent standard deviation from the mean. Panel a was created with BioRender.com. VTM viral transport medium, SDS-PAGE sodium dodecyl sulfate-polyacrylamide gel electrophoresis, BEETLES[2] bioengineered enrichment tools for the LFA with enhanced sensitivity and specificity.

pressure was 3 bar. With a similar molecular weight, enrichment only with N proteins was approximately 24.16-fold, whereas BSA showed no significant enrichment. Unlike the negatively-charged BSA's tunable properties between enrichment and separation with pressure, positively-charged N proteins showed only enrichment without tunability. These results indicate that the BEETLES[2] membrane completely blocks the penetration of positively charged proteins through the filters via the electrostatic interaction of negatively charged phospholipids and positively charged viral proteins under physiological conditions[37]. The enriched N protein samples were acquired after 3 min of BEETLES[2] operation (SDS-PAGE in Fig. 3e). In contrast, no enrichment from BSA in the enrichment zone was observed after 3 min.

To check the surface charge effect on enrichment, representing permselectivity, we controlled the surface charge of BSA (Fig. 3g). We conducted A280 absorbance measurements using NanoDrop to define the enrichment factor. Under a pH condition below the pI value of BSA (pI: 4.5–4.8), the overall surface charge was positive. At pH 3–4, under positively charged conditions, BSA was enriched up to 18.2-fold. Interestingly, the enrichment factor abruptly decreased near the pI value, where the overall charge seemed neutral. The most likely explanation for enrichment is the electrostatic force between the positively charged protein and negatively charged BEETLES[2] membrane. At higher pH, the overall charge of BSA was negative. Thus, no significant decrease in the enrichment factor was observed.

**Enhanced assay performance using the BEETLES[2] membrane**
Figure 3h shows the enhanced sensitivity of COVID-19 Ag using BEETLES[2]. Compared to a commercial LFA (AllCheck COVID-19 Ag, Calth Inc., Republic of Korea) without BEETLES[2] enrichment, all colorimetric signals via BEETLES[2] increased, indicating higher detection sensitivity. We prepared seven different concentrations from by diluting N

proteins with a PBS buffer and tested three samples for each point. The LOD was enhanced up to 20-fold after BEETLES[2] enrichment.

To validate the versatility of the COVID-19 assay, we conducted the COVID-19 IgG rapid test (Fig. 3i). Similar to the COVID-19 Ag test, we observed enhanced sensitivity using the sample enrichment process. The test was performed thrice per point for seven different concentrations. Consequently, the LOD was enhanced up to 20-fold after BEETLES[2] enrichment, showing great applicability in the LFA assay. We confirmed that BEETLES[2] could enrich N proteins and IgG, which are the primary targets for the SARS-CoV-2 test. Moreover, it is likely that BEETLES[2] could be applied to other target viruses, such as Influenza A/B since their N proteins and IgG properties are similar to SARS-CoV-2. To validate the applicability of BEETLES[2] to other viruses, we performed an Influenza rapid test (Fig. S10). We used Influenza Antigen A (Shangdong/9/93(94/516, NIBSC, United Kingdom) and B (Wisconsin/1/2010 (cell derived) (12/110, NIBSC, United Kingdom) with their rapid kits (SGT i-flex Influenza A&B, Sugentech, Republic of Korea). Similar to the COVID-19 Ag test, we acquired an enhanced colorimetric signal using the sample enrichment process.

To demonstrate salivary assay feasibility, we performed the COVID-19 Ag assay using saliva samples (Fig. 3j). The N proteins spiked in artificial saliva for seven different concentrations. As commercial LFA was not optimized for salivary assay, the colorimetric data with commercial COVID-19 Ag LFA revealed lower sensing performance

with higher LOD. With BEETLES[2], we increased the sensing performance with increasing LOD up to 20-fold.

### Enhanced clinical assay using the BEETLES[2] membrane

Clinical samples (COVID-19 patients ($n = 42$) and healthy controls ($n = 20$)), including NP/OP swabs ($n = 30$) and saliva samples ($n = 12$), were used to validate BEETLES[2]'s clinical potential (Fig. 4). To clearly demonstrate performance enhancement, we included samples with intermediate ($26 \leq Ct < 30$, $n = 15$: 15/42, 35%) and low viral loads ($Ct \geq 30$, $n = 20$: 20/42, 48%) where the sensitivity of commercial LFA abruptly dropped.

Reportedly, the test accuracy must be understood for different groups and samples because of the heterogeneity of antigen tests and population groups[8]. We analyzed the most widely spread SARS-CoV-2, Delta and Omicron variants. Reportedly, the pooled percentage of asymptomatic infections was 32.40% for Omicron[38]. We also checked the ability to test asymptomatic individuals. Most samples were collected at Seoul St. Mary's hospital. We collected clinical NP/OP in a viral transport medium (VTM) and assay prepared using the standard protocol according to the LFA manufacturer's guidelines. We called the commercial LFA for control as *w/o BEETLES*[2] and denoted BEETLES[2]-assisted commercial LFA as *w/ BEETLES*[2]. We acquired the cut-off value by maximizing the sum of sensitivity (true positive) and specificity (true negative) of COVID-19 tests.

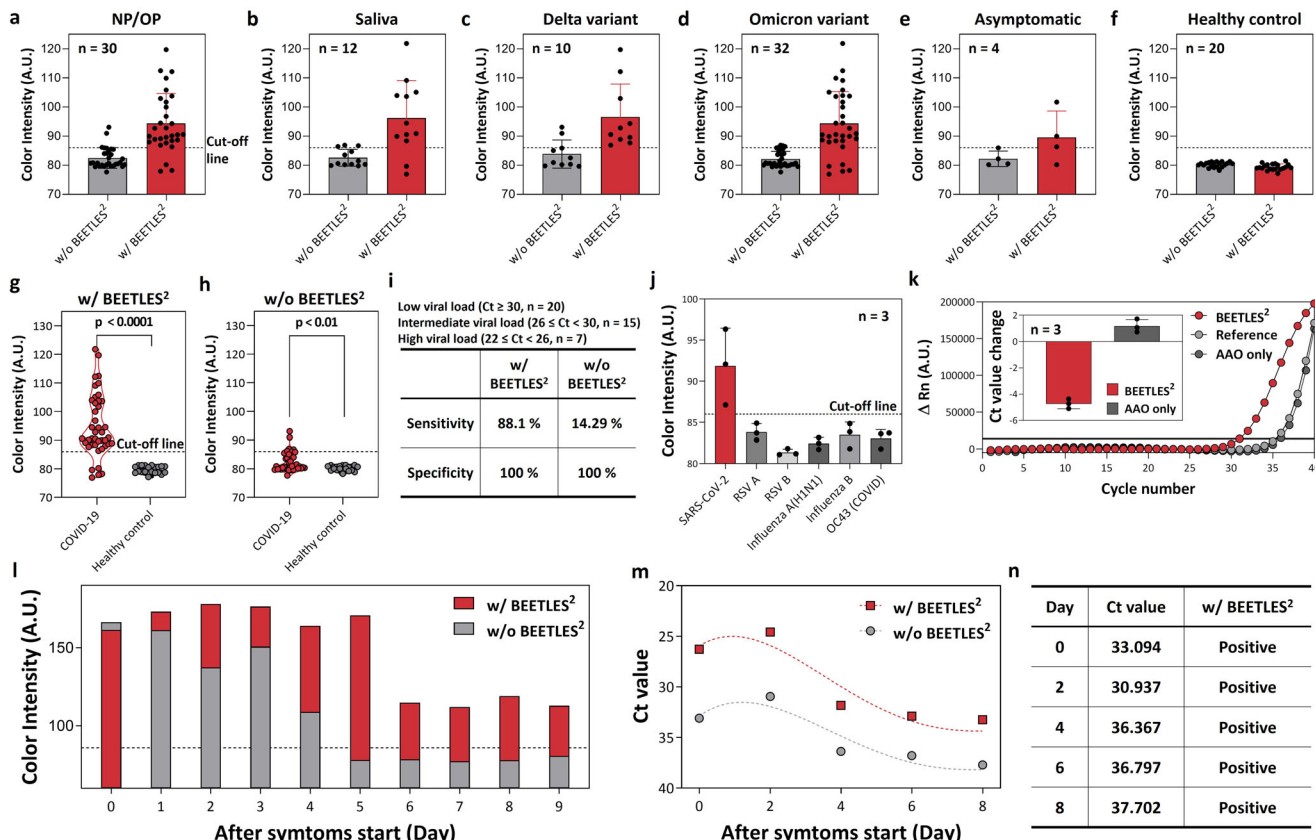

**Fig. 4 | Clinical performance validation.** Clinical samples include COVID-19 patients ($n = 42$) and healthy controls ($n = 20$). We included low-viral patient samples (Ct ≥ 30, $n = 20$: 48%) to observe the enhancement in the case of low viral load. We performed the clinical test for each patient once ($n = 1$). **a–f** Sensitivity enhancement with BEETLES[2] using (**a**) NP/OP samples, (**b**) saliva samples, (**c**) Delta variants, **d** Omicron variants, (**e**) samples from asymptomatic individuals, and (**f**) healthy controls. **g–i** Diagnostic accuracy of BEETLES[2], including all the samples of NP/OP, saliva, asymptomatic, and various variants, shows that the overall sensitivity of BEETLES[2] is significantly higher (88.1%) ($P < 0.0001$) than those using commercial LFA (14.29%) ($P = 0.0041$). Two-sided unpaired t-test was used for analysis and

adjustments were not made for multiple comparisons. **j** Cross-reactivity using different respiratory viruses showing no cross-reactivity. **k** RT-qPCR with enrichment with AAO only and BEETLES[2], showing that BEETLES[2] can enhance molecular diagnostics even with VTM. **l–n** The feasibility of daily monitoring of COVID-19 represents the ability of frequent tests with significantly higher accuracy. With BEETLES[2], we measured the Ct value over 37. Error bars represent standard deviation from the mean. VTM viral transport medium, BEETLES[2] bioengineered enrichment tools for the LFA with enhanced sensitivity and specificity, AAO anodic aluminum oxide, LFA lateral flow assay, NP/OP nasopharyngeal/oropharyngeal.

First, we checked the sensitivity enhancement with BEETLES[2] using NP/OP samples (Fig. 4a: $n = 30$). After loading, samples from NP/OP were diluted with a PBS buffer and injected into the stirred cell. A constant pressure (~3 bar) was then applied with a regulator. Generally, the sensitivity of commercial LFA abruptly decreased virus titers in intermediate and low viral loads (Ct ≥ 26). With NP/OP samples, the sensitivity of commercial LFA was 10% (3/30 = 10%). The sensitivity was then increased with BEETLES[2] (27/30 = 90%).

Second, we validated sensitivity using saliva samples (Fig. 4b). An assay using saliva samples has significant advantages for easy and non-invasive sample collection[39]; however, salivary assays need additional sample collection steps such as prefiltration. For the prefiltration of cells and debris, we used a 450-nm filter membrane. Reportedly, tests using saliva samples have higher detection sensitivity in the first 10 days after infection[40]. Compared to the commercial LFA's sensitivity before enrichment (3/12 = 25%), all the samples tested using BEETLES[2] had higher sensitivity (10/12 = 83.3%), indicating performance enhancement postsample enrichment. For the practical use of salivary assays with BEETLES[2], we suggest a simple saliva collection and purification system (Pure-SAL, Oasis Diagnostics) (Fig. S11a).

Third, we observed the sensitivity using two variants (Fig. 4c, d, respectively). Generally, the LFA of the COVID-19 Ag test targets N proteins. Unlike RT-qPCR, which strongly depends on the variant, such as Delta and Omicron[41], the N proteins targeted by the COVID-19 Ag test are less dependent on mutations. Both for Delta ($n = 10$, Fig.4c) and Omicron ($n = 32$, Fig. 4d), all the samples were tested using BEETLES[2] and commercial LFA. Consequently, we acquired higher sensitivity for Delta (10/10 = 100%) and Omicron (27/32 = 84.38%) with BEETLES[2] than with commercial LFA: 20% (2/10 = 20%) and 12.5% (4/32 = 12.5%) for Delta and Omicron, respectively.

Fourth, we considered asymptomatic cases (Fig. 4e). Fast identification of asymptomatic cases is a key strategy for reducing onward transmission of COVID-19 viruses[42]. The true positive values, i.e., sensitivity, using BEETLES[2] was higher (3/4 = 75%) than the commercial LFA (controls: 0/4 = 0%). We elucidated data points near the cut-off line with LFA test results (Fig. S12). Although the number of asymptomatic samples was not sufficient, BEETLES[2] might provide earlier and better detection of asymptomatic individuals.

Fifth, to check the specificity, we tested the colorimetric signals from healthy controls ($n = 20$) showing 100% specificity (true negatives = 20/20) (Fig. 4f). Specificity is known as the true negative rate. Therefore, we increased the specificity by reducing false positives.

Overall, we assessed the diagnostic accuracy of BEETLES[2], considering all samples of NP/OP, saliva, asymptomatic, and various variants (Fig. 4g and h). Figure 4g shows the assay with BEETLES[2], and Fig. 4h shows the assay with commercial LFA. The overall true positive values, i.e., sensitivity, using BEETLES[2] is significantly higher (37/42 = 88.1%) than those using commercial LFA without BEETLES[2] (6/42 = 14.29%) (Fig. 4i). Our proposed BEETLES[2] system meets the World Healthcare Organization's (WHO) 'desirable criteria' guidelines for POCT (sensitivity: >90%, specificity: >99%, Ct ≥ 30, and assay time: <20 min) beyond the 'acceptable criteria' (sensitivity: >80%, specificity: >97%, 26 ≤ Ct ≤35, and assay time: <40 min)[43].

Another important parameter for COVID-19 diagnostics is cross-reactivity (Fig. 4j). We evaluated cross-reactivity using different respiratory viruses such as respiratory syncytial virus A (RSV A), RSV B, Influenza A(H1N1), Influenza B, and Human coronavirus-OC43 (HCoV-OC43) ($n = 3$ for each sample) and observed no cross-reactivity.

Both the LFA test and BEETLES[2] enrichment could contribute to cross-reactivity and specificity. However, since the LFA manufacturer strictly controlled cross-reactivity and specificity (>99%), we expect that the results of cross-reactivity and specificity are primarily due to BEETLES[2] enrichment (Table. S1). We observed no cross-reactivity and specificity problems for other viruses, demonstrating a high potential to meet WHO criteria.

An AAO membrane has been used to develop an immunoassay for SARS-CoV-2 detection in saliva by enriching SARS-CoV-2 viruses[9]. Following that study, we prepared only an AAO membrane and observed virus purification and enrichment via RT-qPCR (Fig. 4k). In PBS buffers, the intact viruses could be enriched through the AAO membrane, as presented. However, we could not enrich virus particles using the AAO membrane in a lysate buffer (VTM) due to a lack of VTM lysates viruses for enrichment. Therefore, the Ct value using the AAO membrane did not change significantly. In many real clinical cases, samples are delivered with inactivated lysate (i.e., VTM). The Ct value was improved using BEETLES[2], even with VTM, up to 5.1, theoretically corresponding to 34.3 enrichment concentrations.

## Sample-to-answer platform for daily monitoring

To achieve the daily monitoring feasibility for COVID-19, we collected daily samples of NP/OP and saliva from infected individuals with symptoms (Fig. 4l–m). The day of onset (day 0) was defined as the first day of symptoms. All the patients were RT-qPCR positive on the first day of symptoms. We acquired daily data via RT-qPCR and LFA with BEETLES[2] enrichment. As RT-qPCR can detect viral shedding long after the infectious period (~9 days), the positive result from RT-qPCR might include unnecessary quarantine. Conversely, frequent highly sensitive LFA testing (2–3 daily tests) can provide changes in the on/off signal, even the quantitative data. Daily monitoring data revealed that the LFA assay with BEETLES[2] is highly correlated with the viral load (Ct value), representing that our technique evaluates frequent tests with significantly higher accuracy than the commercial LFA and RT-qPCR. With BEETLES[2], we could measure the Ct value over 37, which the commercial LFA could not detect (Fig. 4n).

Compared with other existing virus isolation and enrichment techniques, our BEETLES[2] has advantages in the permselectivity, tunability, powerful enrichment capability for both intact viruses and their N proteins, and applicability as a hand-held device (see Table. S2 for the comparison of different virus isolation and enrichment focusing techniques). To commercialize BEETLES[2], we should address the following issues: 1) pressure endurable design. Since the BEETLES[2] assay is performed under applied pressure, no liquid sample leakage can be tolerated. Especially when handling infectious samples, a leak-free design is needed for safety. 2) The volume issue in serological rapid diagnostic tests. For serological rapid diagnostic tests, 1–2 drops (<100 μL) a typically acquired from a finger prick; therefore, we need to develop devices for blood-based tests with small sample volumes. A microfluidic device integrated with the BEETLES[2] membrane is an ideal candidate for serological assays, 3) Shelf life is a key parameter to commercialize BEETLES[2]. We obtained stability, for up to 15 days, under room temperature. However, the expected shelf life can be extended further with additional tests (Fig. S13).

In conclusion, BEETLES[2]'s sample preparation with its powerful permselectivity and tunability provides more sensitive binary response than commercial LFA in 3 min. It has two significant practical advantages: first, the system can be easily coupled with well-established commercial LFAs and increase its clinical sensitivity (>88%) and accuracy (>91%) for low titer clinical samples including Delta and Omicron variants, compared with commercial LFA (14.29% and 41.94%, respectively). Second, BEETLES[2] can selectively enrich the SARS-CoV-2 virus, N proteins, and IgG from the nasal swab, saliva, and blood serum, showing its versatility and applicability to COVID-19 Ag and IgG testing.

Combining sample enrichment of BEETLES[2] with commercial LFA represents an advance in POCT detection since it meets all the requirements stated by WHO for POCT, i.e., ease of use, low cost, and accuracy[44]. We expect BEETLES[2] to be commercialized at a nominal additional cost, thereby meeting the inexpensive criteria. Moreover, the combination of AI-assisted smartphone applications with BEETLES[2] could precisely predict on/off switching and perform quantitative classification, demonstrating great potential for the REASSURED (Real-

time connectivity, Ease of specimen collection, Affordable, Sensitive, Specific, User friendly, Rapid, Equipment free and Delivered)[45], a new criterion for digital connectivity, and possibly detect asymptomatic transmission with simple frequent test strategies[8,42], which is difficult with RT-qPCR testing.

## Methods

### Ethical statement
Respiratory samples were prospectively collected from patients diagnosed with COVID-19 infection at Seoul St. Mary's Hospital from April 2021 until May 2022. This study was approved by the institutional review board (KC21TIDI0134K) at Seoul St. Mary's Hospital, and informed consent was obtained from the participants.

### BEETLES[2] membrane: preparation of RBCM
We extracted RBCM from K2EDTA anticoagulated human whole blood and removed the plasma and buffy coat using an 800-g centrifuge for 5 min to isolate red blood cells. After removing the supernatant, we washed the remaining red blood cells thrice with ice-cold 1×PBS with gentle handshaking. Next, we hemolyzed the red blood cells by suspending them in ice-cold 0.25×PBS for 30 min. Finally, we eliminated free hemoglobin using a 20,000-g centrifuge for 30 min. After three wash-outs, a light pink RBCM pellet was obtained and stored at −80 °C for future use. The extracted RBCMs were in the form of a negatively charged liposome of 200 nm size (Fig. S2). The sonicated solution was dropped on the AAO membrane (6809-6002, Whatman, UK) and incubated for 30 min at 50 °C to form multiple supported lipid bilayers. To investigate the microstructure of the BEETLES[2] membrane, we conducted structural analysis using SEM, AFM, and fluorescent microscopy.

### BEETLES[2] membrane: property analysis
AAO-RBCM membranes were sampled for KPFM imaging on a p-type silicon wafer (ePAK International, USA), a substrate with electrical conductivity. The silicon wafers were immersed in piranha solution ($H_2O_2$ and $H_2SO_4$) for 15 min, washed with distilled water, and dried with $N_2$ gas (Sejong Industrial Gas Co., Korea). A MultiMode VIII atomic force microscope was used to analyze the AAO-RBCM membranes electrically and topologically in amplitude-modulated KPFM mode (Bruker, USA). KPFM measurements were performed at 23 °C in air using the lift scan mode based on the tapping mode. Conductive AFM tips coated with Pt (SCM-PIT-V2; Bruker, USA) were used to analyze the nano-electrical properties of the samples. A topological AFM image was captured in the first scan using the tapping mode with a zero-tip bias. To detect the surface potentials, the AFM tip was lifted 20 nm above the sample surface with an applied sample bias voltage during interleave scanning. The mechanical drive to the cantilever was turned off during interleave scanning, and a bias voltage of alternating current (AC) at 1000 mV was applied to the probe at the mechanical resonance (ω) of the cantilever. The $V_{AC}$ causes the cantilever to oscillate owing to attractive and repulsive electrostatic interactions ($F_{es}$) between the probe and sample, defined as follows:

$$F_{es} = -\frac{1}{2}\frac{dC}{dz}[(V_{DC} - V_{CPD}) + V_{AC}\sin(\omega t)]^2 \qquad (1)$$

where $V_{DC}$ is the direct current (DC) bias voltage, and $V_{CPD}$ is the contact potential difference between the probe and sample.

Applying a compensating $V_{DC}$ to the probe to eliminate electrostatic forces (i.e., $F_{es}$) between the probe and sample establishes a proportional-integral-derivative feedback loop that monitors and controls the amplitude of cantilever oscillations. These depend on the capacitance C and height z of the probe and sample. The KPFM scan rate was 0.6 Hz, the scan size and amplitude setpoint at 12 nm was 4 μm × 4 μm at 512 × 512 pixels, respectively. The MountainsSPIP

software was used to level, process, and analyze the AFM images (version 9; Digital Surf, France). In addition, using Gwyddion, surface roughness studies were carried out (version 2.60).

FRAP was used to validate the SLB formation of RBCMs for assessing their fluidity and diffusivity. In all experiments, 1 wt% PE-CF was intercalated into RBCMs as the fluorescent probe. We bleached a 20 μm diameter spot at the z-plane of the RBCM layers via a 488-nm optically pumped semiconductor laser (Coherent, Inc.) at 150 mW for 5 s. Fluorescence recovery after photobleaching at the desired spot was monitored for 5 min via a Zeiss LSM-800 confocal microscope. Each image reference spot served as a standard for measuring and normalizing spot fluorescence intensity. The normalized fluorescence intensity against time fit with a Bessel function was reported by Soumpasis et al.[26]. Next, the diffusion coefficient (D) of the dye in each SLB composition was calculated using the following expression: $D = w^2/4t^{1/2}$, where w is the radius of the photobleached spot and $t^{1/2}$ is the time required to achieve half of the maximum recovery of fluorescence intensity.

### Assay: Reagents and analysis
We used commercial COVID-19 Ag rapid kits (Calth Inc., Republic of Korea) and tested the enhanced performance via BEETLES[2]. We prepared a COVID-19 N protein recombinant antigen (45 kDa, FPZ0516, Fapon Biotech Inc., China), considered the best COVID-19 Ag test target, using 1×PBS buffers (LB004, DUKSAN, Republic of Korea) to test sensitivity and LOD. Bovine serum albumin (BSA) (A7030, Sigma-Aldrich, USA), Human IgG (I4506, Sigma-Aldrich, USA) and Human IgG Fc fragments (401104, Sigma-Aldrich, USA) were purchased from Sigma-Aldrich. Subsequently, we prepared seven concentrations of the COVID-19 N protein sample with 1xPBS. We added ~3 mL of each sample onto a polycarbonate stirred cell (341000, STERLITECH, USA) and operated for 3 min. Next, the 100 μL of the enriched sample was recovered with a pipette. NanoDrop spectrophotometers (Thermo Scientific, USA) and SDS-PAGE (KOMA precast gel BC type, Koma Biotech, Republic of Korea) were used to analyze and quantify proteins.

For the COVID-19 Ab test (Fig. 3i), we prepared seven concentrations of the COVID-19 antibody (sodium citrate plasma, Trina Bioreactives AG, Switzerland) sample with 1xPBS. We added 3 mL of each sample into a polycarbonate stirred cell (341000, STERLITECH, USA) mounted with an RBCM-coated AAO membrane and operated it for 3 min. Next, the enriched sample was recovered via a pipette with 100 μL LFA extraction buffer. Subsequently, we used the enriched sample in the COVID-19 Ab LFA kit (AllCheck COVID-19 IgG/IgM, Calth Inc., Korea) and noted the result at 10 min.

Similarly, for the salivary COVID-19 Ag test (Fig. 3j), we diluted seven concentrations of the COVID-19 N protein sample with artificial saliva (A7990, Solarbio, China). First, we added 3 mL of each sample into a polycarbonate stirred cell mounted with an RBCM-coated AAO membrane and operated it for 3 min. Next, the enriched sample was recovered via a pipette with 100 μL of LFA extraction buffer. Subsequently, we used the enriched sample in the COVID-19 Ag LFA kit and noted the result at 10 min.

We analyzed the color intensities of the commercial LFA kit and BEETLES[2]-assisted LFA using a custom-made National Instrument (NI) controlled optical system coupled with LabVIEW v 2019 SP1 (National Instruments Co., USA). To set the LOD (Fig. 3) and cut-off values (Fig. 4), we first measured color signals using the commercialized reader and custom-made NI-controlled optical system with LabVIEW. Next, five individually trained engineers (Calth Inc. http://www.thecalth.com) observed the colorimetric signal using the standard color chart and manufacturer's guidelines to determine the LOD. Meanwhile, we set the cut-off values to minimize false positives for acquiring higher specificity under the manufacturer's guidelines.

## Assay: clinical validation

We collected clinical samples of NP/OP and saliva from COVID-19 patients at Seoul St. Mary's hospital with appropriate Institutional Review Board Committee approval (KC21TIDI0134K). Although NP/OP swabs were contained in the viral transport medium (VTM), saliva samples were collected in a sterilized tube and diluted with PBS.

To test the performance in low-viral targets, for high viral load (Ct <26), we spiked NP/OP samples into PBS buffers and prepared them as low-viral samples. For saliva samples with a high viral load, we spiked the samples with PBS buffer. Finally, we prepared low-viral samples with Ct ≥ 30. For healthy controls, we collected (or purchased) samples from healthy volunteers and stored them at −20 °C. The obtained results were compared with RT-qPCR SARS-CoV-2 tests using Accu-Power® SARS-CoV-2 variants ID2 kit (BIONEER, Republic of Korea), according to the manufacturer's protocol. The cut-off value for on-site COVID-19 diagnostics was acquired by maximizing the sum of sensitivity (true positive) and specificity (true negative) of COVID-19 tests.

To realize the prototype for POCT sample preparation, we fabricated a hand-powered portable gadget integrated with the BEETLES[2] membrane (See Fig. S11). We designed two reservoirs: sample and commercially available running buffer reservoirs. Micro milling was used for the prototype. We calculated the average hand-powered pressure from five individuals (4 men and 1 woman) with BEETLES[2] as 2.3 ± 0.2 bar (Fig. S14). We showed that the permselectivity of BSA and N proteins was maintained under hand-powered pressure. Hand-powered portable syringes are low-cost alternatives for point-of-care diagnostics, which can be directly applicable to commercial LFA. We first connected the BEETLES[2] membrane to the sample reservoir and pushed the plunger for enrichment. After 3 min of operation, we reversed the direction of BEETLES[2] membrane attachment to the running buffer reservoir. Since the commercial running buffer included a surfactant, we could successfully demonstrate the N protein assay regardless of the electrostatic force between the N protein and RBCM layer. Subsequently, we assayed using commercial LFA following the manufacturer's guidelines. With further investigation, we are developing a simplified second prototype wherein a vacuum chamber is used for applying pressure.

## Statistics and reproducibility

The error bars presented in the figures represent the mean ± SD. We repeated all the experiments at least thrice per point and analyzed the data using Microsoft Exel, Prism v 7.0. Biorender, 3D MAX v 2020, V-Ray v 4.0, Adobe Photoshop v 2020, and Adobe Illustrator v 2020 software were used for graphical analyses. No statistical method was used to predetermine sample size. To determine the significance of the clinical validation, we analyzed its data with a two-sided unpaired $t$-test to examine the differences between the groups. A $P$ value below 0.05 was considered statistically significant.

## Data availability

Source data are provided with this paper.

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

## Acknowledgements
This research was supported by the Bio & Medical Technology Development Program of the National Research Foundation (NRF) funded by the Korean government (MSIT) (No. 2021M3E5E3080743).

## Author contributions
S.J.P., S.L., D.L. (Dongtak Lee), D.S.Y., Y.K.Y., and J.H.L. conceived and designed the study. N.E.L., J.S.P., and J.H.H. conducted the PCR experiments. D.L. (Dongtak Lee), J.W.J., and H.K. performed the extraction of the red blood cell membrane and material characterization. S.R. and G.L. conducted Kelvin probe force microscopy-based material characterization. D.L. (Dongho Lee), S.C., C.P., D.L. (Dong-Gun Lee), R.L., and D.N. provided clinical samples and interpreted the results. S.J.P., S.L., D.L. (Dongtak Lee), Y.K.Y., and J.H.L. drafted the manuscript. All authors discussed the results and commented on the manuscript.

## Competing interests
The authors declare no competing interests.
