## [Peer Review File · Nature Communications]

REVIEWER COMMENTS

Reviewer #1 (Remarks to the Author):

Here are my comments:

General comments:

The team described the bioengineering of an enrichment tool (BEETLES) which combined with COVID-19 LFA allows the enhancement of analytic performance for nucleocapsid (N) proteins and/or immunoglobulin G (IgG) detection. This is an interesting and unique study, but has several limitations that I would invite the authors to consider.

Specific comments:

Line 67: LFAs provide convenient, easy-to-use, one-step, disposable, and fast results with 108 entities per milliliter. The sentence is not clear. Please precise "entities"

Line 100: The use of smartphones to improve on the one hand the reading of the results of rapid diagnostic tests and on the other hand, the transmission of the results to a laboratory computer system is a well-known subject. This point is not in my eyes the heart of the article and brings little added value to the description of the BEETLES.

Line 116-117: "However, we observed the tunable permselectivity with applied pressure". Unclear what is meant by "Pressure". Please clarify by giving the range.

Line 177-179: Authors stated that they enriched 3 mL of each sample into a final volume of 100 μ L, theoretically corresponding to 30-fold enrichment. Most serological rapid diagnostic tests require a much smaller volume of blood, around 50 to 100 μ L (one to two drops of blood). This point must be discussed because it completely changes the possibility of using this bedside test in a POCT perspective.

Line 201: Even if the name of the LFA is briefly described in the material and method, please specify the manufacturer name of the LFA but also its performance (Se and Sp).

Line 212: The BEETLES could be interesting for other respiratory viruses. Could the authors clarify if this could also be useful for RSV detection?

Line 220: Please precise the number of NP/OP swabs and saliva samples tested.

Line 234: Phrasing " We optimized the cut-off value for on-site COVID-19 diagnostics by maximizing the sum of sensitivity and specificity " is not clear. Revise

Line 239: Please express the viral load in copies/mL instead of Ct.

Line 242: I do not agree with the following sentence "Assay using saliva samples has great advantages in easy sample collection". The collection of saliva is not always easier because you have to salivate a lot to produce the necessary volume. This is seen especially in symptomatic patients. The use of sponge solved this problem. It should be explained how the authors envisage the use of BEETLES following this type of sample (sponge).

Line 265: In the following sentence "Our system meets World Healthcare Organization (WHO)...", please precise the term our system.

Line 269: What is meant by "selectivity" here-sentence confusing?

Line 275: Phrasing " By reducing false positives, " is not clear. Revise

Line 278: The following sentence "Chen et al. reported a quantitative and ultrasensitive in situ immunoassay using an AAO membrane, showing the ability to enrich SARS-CoV-2 viruses" should be placed in Line 77 when authors introduced AAO.

Line 299: Please add a sentence describing the benefit of smartphone connection by describing more in depth REASSURED criteria, which are only mentioned in line 316.

Line 313: The authors should explain how they figure an additional cost of tens of cents

Figure 1: Please clarify the abbreviation “BSA”

Figure 1C: The use of smartphones to improve on the one hand the reading of the results of rapid diagnostic tests and on the other hand, the transmission of the results to a laboratory computer system is a well-known subject. This point is not in my eyes the heart of the article and brings little added value to the description of the Beetles. It would be wise to delete part C of figure 1. Nevertheless, discuss a possible connection of the quick test at the end of the chapter finding before the conclusion.

In addition to the edition of these various points, the authors should also add a paragraph describing how they evaluated the potential leaks of liquids through the BEETLES device. This is extremely important because the notion of User-friendly taken up in the REASSURED criteria also includes the safety of the user, especially when the latter handles potentially infected samples such as respiratory samples or blood.

Conclusion:

With attention to the above details, the paper is worth to be published.

Reviewer #2 (Remarks to the Author):

Paper 1: PCR-like Performance of Rapid Test with Permselective Tunable Nanotrap

Summary

The manuscript presents a nano trap with interesting permselectivity that can be used with commercial LFA. They use an RBCM lipid membrane on top of the AAO membrane to filter and enrich before using LFA. It reports bioengineered tools comprising nanoporous AAO membranes functionalized with red blood cell membranes to improve the antigen/antibody for the lateral flow assay with enhanced sensitivity and selectivity. The reported tools can eliminate the interfering proteins and increase the targets with a 20-fold enhancement of the detection limit. Clinical samples with different sample types and variants are tested using the proposed tool, validating its practical applications for a rapid diagnostics test. Overall, the authors claim to compete with PCR in terms of performance. The clinical claims of the manuscript are promising. However, some significant revision comments need to be addressed.

Major Comments

1. The manuscript and the reported device are well characterized. I can predict that the reported device shows great potential for practical application for home diagnostics. However, a similar mechanism has been reported (Chen et al. Quantitative and ultrasensitive in situ immunoassay technology for SARS-CoV-2 detection in saliva. *Sci. Adv.* 8, eabn3481 (2022).) although the addition of red blood cell membranes can help to enrich the specific targets. Please show a competitive analysis concerning many similar works and justify why it has authentic merits.

2. Page 6, Line 128, "The optimal RBCM concentration was checked by AFM .." – It is mentioned that AFM is the deciding factor on optimal thickness. It is not clear how AFM observation can justify optimal thickness. In AFM images, even a 1 % sample seems pretty covered in AFM but not in SEM. Fig. 2d maybe not be necessary in that case. It would be helpful to mention the thickness with error %.

3. Fig. 3f, Why do KPFM measurements show a reduced magnitude of negative charge for the 4% RBCM sample? How significant is the importance of potential for achieving the desired permselectivity?

4. There seems to be a drastic change in RBCM deposition from 1 % to 2% compared to other consecutive pairs (SEM images, Fig. S3). The samples of 0.5 % and 1 % seem very similar. Please justify the reason. If 2% is the first sample to be fully covered, it would be ideal to choose a % slightly above that as optimum.

5. Elaborate on the novel functionality that the RBCM membrane adds to the filter. If the 20 nm AAO membrane is used as-is, size-based filtration is possible. But that might cause nucleocapsid protein to escape if there are lots of them in the saliva sample. It will be helpful to describe how many direct nucleocapsid proteins can be there in the saliva sample to justify the RBCM layer.

6. Small syringes can have a much larger pressure than 3 bar; even Fig. S8 shows a 6-bar pressure. Fig. 3b shows that increasing pressure can tune the enrichment. It helps to remove BSA, but it can cause NP to de-enrich. There is a trend of de-enrichment of NP in Fig. 3f as pressure increases. What is the possibility that increased pressure might allow nucleocapsid protein to pass through, overriding the electrostatic interaction between NP and RBCM? The system may be very pressure sensitive in normal pressure ranges. It is important to study that aspect.

7. Fig. 3b, 3d, 3f, 3g- Is the effect of pressure on enrichment far more overriding than other aspects-charge and pH? It is necessary to increase the pressure window and determine if charge and pH are still making a significant difference.

8. Page 2, line 34- The tunability of nanotrap is one of the claims. Enrichment tunability under different pressures is a standard principle used in filtration. There is no new insight added to that tunability in the manuscript. Hence can be avoided as a claim.

9. Page 16, line 421- While transferring the enriched sample to LFA, the electrostatic force should still make the NPs attached to the RBCM layer, although it is inverted. How is that overridden? It should be explained. How much pressure is applied while LFA application?

10. Explain why Fig. 4f shows no cross-reactivity.

11. How is the long RBCM layer stable at room temperatures? A study about the storage of the final membrane outside can supplement the analysis.

Minor Comments

1. Fig. 2e and 2f convey similar information, and one can be avoided.
2. Please number supplementary sub-figures.
3. Please increase the size of the text inside the figures.
4. Fig. 2g and 2h convey similar information, and one can be removed.

Conclusion

The manuscript presents exciting ideas. However, the enrichment, tenability, selectivity, and specificity observations should be explained more scientifically. The choice of RBCM concentrations should be defined more convincingly. The pressure range that they tested the system needs to be evaluated. After a critical major revision, the manuscript can be considered.

Reviewer #3 (Remarks to the Author):

This paper reports the development of an LFA tool for sensitive and selective COVID-19 detection. This method uses a perm-selective tunable Nanotrap to enhance clinical testing performance for infectious agents. The results described purport the enablement of more accurate and rapid point-of-care testing, and for a lower cost than RT-qPCR. The narrative should not convey that the BEETLE system allows for results comparable to RT-qPCR, as LFA (antigen) tests will always have drawbacks associated with a passive flow microfluidics, sensitivity and visual detection, however, the ability to rapidly enrich samples using the perm-selective Nanotrap system certainly aids sensitivity and could enhance clinical performance.

Overall, the paper is well written, although it is not conversational – the narrative unclear and somewhat ‘dry’ to this reviewer – but this could simpl be stylistic differences. The figures are detailed and easily digestible, however, resolution could be improved. The supplemental video was very helpful.

Major Issues:

The intro provides a good overview of the motivation for the project, but it does not introduce much about the method selected for COVID-19 enrichment. For a more complete introduction, it should

provide some background description of the tunable Nanotrap (prior use, 'mechanism' (how it works, etc.). Additionally, while it is appropriate to discuss the 'WHO criteria' for POCT (actually, multiple times at the end of the paper), it would be beneficial to discuss these in the introduction as well.

With the main findings, it's not clear exactly how the enriched fractions of samples are being further processed for analysis and subsequent detection. While this is clarified somewhat by the video, the body of the text should describe the unit operations associated with handheld device. This would nicely compliment moving supplemental figure 7 to the main body of text for ease of visualization.

There is a lack of clarity as to which buffer is used and what the context is for use of the different range of buffers used in these types of applications and how this aids in assay flexibility. [Page 5 (lines 101 – 102)]

The definition of "tunable" is not clear from the descriptions provided. The authors need to include more text describing exactly what is being 'tuned' and how that can be applied to different analytes. [Page 6 (lines 115 – 117)]

From the text, it appears that several studies were conducted to determine the optimal RBCM concentration, and it seems the 2% concentration was based on KPFM and fluorescent data alone. If this is the case, either provide an explanation to describe why all other data was excluded, or link this data back to the other experiments (Figs 2d, s3, s4, 2e – f, and s5, s6). [Page 6 (lines 136 – 137)]

Figure 3 – When testing patient samples, was each sample split into 3 aliquots to run with and without BEETLES2? While I recognize that the spread of data points is due to patient-dependent differences in viral load, stochastic differences between aliquots of this patient sample would be noteworthy data points. If this was not done, clarify that this is an n = 1 for each patient sample.

The verbiage associated with Figure 4e states that with the BEETLES2 system, $\frac{3}{4}$ of true positives were called, whereas the commercial assay called 0/4. However, it appears from the figure that each condition has one sample residing on the threshold line, and it was called in favor of the manuscript narrative. This needs to be addressed.

Please comment on whether the observation of "no cross-reactivity" was a result of the specificity of the downstream LFA test, or the result of permselectivity of the enrichment process described here. [Page 10 (lines 269 – 273)]

With respect to the 'cost per test' as described here, the authors need to provide details as to how they arrived at those numbers (e.g., would this be at low-rate production? high rate production)? and how easily these membranes could be manufactured.

Minor issues:

- The abbreviation "NP" is used to describe both Nasopharyngeal swabs and N proteins – distinguish.

- Page 3 (Lines 45 – 47) Phrasing needs adjusted as the point is not clearly stated as is.

- Figure 1B & 1C are repetitive aside from Ag vs. Ab test – either combine or remove 1C

- Page 5 (lines 102 – 108) We understand that the pore sizes are tunable based upon pressure; however, it would be advantageous to have some size range described in nm

- Page 9 (line 220) – Create an abbreviation for NP/OP swabs prior to use

- Page 9 (lines 229 – 232) – These sentences describing IRB approval and collection/protocols should be moved to the methods section

- Page 9 (lines 232 – 233) – Make clear that you are naming your results here by using either quotation marks and/or italicized names

- Figure 3a – f – add a sentence to the body of the text to describe why there is such a large spread of LFA results (e.g., sample – to – sample variation from patient viral load)

- Page 10 (line 253) typo – "that" should be "than"

- Figure 4k – please add fluorescence units to the y axis of the qPCR plot and a threshold line for better visualization of Ct value differences

- Figure 4I – Figure very difficult to interpret – consider either splitting this into two bar graphs or explaining how to read this in the figure legend

- Page 12 (line 309) – This conclusion alleges that blood serum samples were tested, but they were not in the body of work described here. Please remove.

Recommendation

This is interesting tech that could be beneficial with scenarios where lost-cost but rapid results are required. I believe this ms could clear the bar for NATCOMMS and, in this reviewer's opinion, resides somewhere between a 'major' and 'minor' revision. But that requires that the issues identified above are adequately addressed.

Reviewer #4 (Remarks to the Author):

The manuscript by Park et al. has developed bioengineered enrichment tools for LFA with enhanced sensitivity and selectivity for rapid detection. The manuscript is well written and can be considered for publication following major revision.

Major concerns:

1. In the KPFM study, with increase in the concentration of RBCM coating the surface potential increased where 2% showed highest followed by a sudden drop on using 4% coating. What could be possible reason for such a sudden drop? Will there be an additive effect if the authors study any percentage between 2-4% coating?
2. For improving permeation, highest tested pressure of about 3 bar was used for the later clinical studies. However, it was observed from Figure 3 b, d, and f, pressure of about 0.5 bar had shown to permeate BSA and able to enrich NP and IgG as like 3 bar. Why have the authors applied 3 bar pressure for later studies?
3. In optimization highest selected pressure of 3 bar has been selected for the further studies, what will be the effect of pressure above 3 bar.
4. AFM topography images, the images could be taken at higher magnification which can give a better idea of changing topography of AAO on RBCM coating.

5. Pressure of 3 bar improved permeation of negatively charged BSA. Is the size of IgG being sole reason for enrichment of IgG or any other properties of IgG or interaction of IgG with RBCM could be plausible for enrichment of negatively charged IgG.

Reviewer #1

Here are my comments:

General comments:

The team described the bioengineering of an enrichment tool (BEETLES) which combined with COVID-19 LFA allows the enhancement of analytic performance for nucleocapsid (N) proteins and/or immunoglobulin G (IgG) detection. This is an interesting and unique study, but has several limitations that I would invite the authors to consider.

answer) Thank you for your feedback on the manuscript. We have revised the manuscript to address these limitations and provided detailed explanations of our methodology and results. We hope that these changes will address your concerns and provide a clearer understanding of our work. We look forward to your further feedback on the revised manuscript.

Specific comments:

Line 67: LFAs provide convenient, easy-to-use, one-step, disposable, and fast results with 10⁸ entities per milliliter. The sentence is not clear. Please precise “entities”

Answer) We mentioned "10⁸ entities per milliliter" in correspondence to reference 22. However, we have deleted that expression in the revised manuscript. We modified the sentence to *"Regarding high-frequency tests for decentralized local testing, the lateral flow assay (LFA) platform is confirmed as the best platform for point-of-care and self-testing as they provide convenient, easy-to-use, one-step, disposable, and fast results²²"* (line 65–67)

Line 100: The use of smartphones to improve on the one hand the reading of the results of rapid diagnostic tests and on the other hand, the transmission of the results to a laboratory computer system is a well-known subject. This point is not in my eyes the heart of the article and brings little added value to the description of the BEETLES.

Answer) As suggested by the reviewer, we have deleted the smartphone part from the “Main Findings section to *"Since BEETLES² can enrich N proteins, SARS-CoV-2 viruses, and IgG*

antibodies, BEETLES²-assisted LFA enables the daily monitoring of infectious diseases." (line 105–106)

Line 116-117: "However, we observed the tunable permselectivity with applied pressure". Unclear what is meant by "Pressure". Please clarify by giving the range.

Answer) We have defined pressure as follows: "*However, we observed tunable permselectivity with an applied pressure >0.5 bar*" (line 124)

Line 177-179: Authors stated that they enriched 3 mL of each sample into a final volume of 100 μ L, theoretically corresponding to 30-fold enrichment. Most serological rapid diagnostic tests require a much smaller volume of blood, around 50 to 100 μ L (one to two drops of blood). This point must be discussed because it completely changes the possibility of using this bedside test in a POCT prespective.

Answer) As the reviewer highlighted, serological rapid diagnostic tests require only 1~2 drops of the sample (<100 μ L). Therefore, we need a new design to address the volumetric issues associated with blood-based tests. We revised the manuscript to "*We enriched 3 mL of each sample into a final volume of 100 μ L, theoretically corresponding to 30-fold enrichment. IgG enrichment was confirmed under a pressure of 3 bar (SDS-PAGE in Fig. 3e). Unlike an antigen or virus-based assay, for serological rapid diagnostic tests, only 1~2 drops (<100 μ L) of blood was used. To handle tens of μ L of samples, a new design is needed. One such prospective design could be the BEETLE2 membrane-integrated microfluidic system.*" (line 197–202)

Line 201: Even if the name of the LFA is briefly described in the material and method, please specify the manufacturer name of the LFA but also its performance (Se and Sp).

Answer) We added the kit information as follows: "*Compared to a commercial LFA (AllCheck COVID-19 Ag, Calth Inc., Republic of Korea) without BEETLES² enrichment, all colorimetric signals via BEETLES² increased, indicating higher detection sensitivity.*" (See line 226–228). The detailed performance sheet of the commercial LFA has been added in *Supplementary Table 1*.

Supplementary Table 1. Performance characteristics of the commercial COVID-19 Ag LFA kit (Sensitivity: 97.5 %; Specificity: 100 %). Ag, antigen; LFA, lateral flow assay.

Clinical performance Result		RT-PCR method		Total
		Positive	Negative	
ALLCheck COVID-19 Ag	Positive	78	0	78
	Negative	2	60	62
Total		80	60	140

Line 212: The BEETLES could be interesting for other respiratory viruses. Could the authors clarify if this could also be useful for RSV detection?

Answer) Since we could not acquire the target samples and rapid kits for RSV, we conducted an Influenza test instead. We revised to *"we can extend BEETLES² toward other target viruses such as Influenza A/B since their N proteins and IgG properties are similar to SARS-CoV-2. To validate the applicability of BEETLES² to other viruses, we performed an Influenza rapid test (Fig. S8). We used Influenza Antigen A (Shangdong/9/93(94/516, NIBSC, United Kingdom) and B (Wisconsin/1/2010 (cell derived) (12/110, NIBSC, United Kingdom) with their rapid kits (SGT i-flex Influenza A&B, Sugentech, Republic of Korea). Similar to the COVID-19 Ag test, we acquired an enhanced colorimetric signal using the sample enrichment process."* (line 237–244).

Supplementary Fig. 10. Enhanced sensitivity assay of Influenza A/B commercial kits via BEETLES². Color intensity of each assay (top) and LFA kit image (bottom). (a) N protein

validation for Influenza A and (b) N Influenza B. BEETLES²: bioengineered enrichment tools for LFA with enhanced sensitivity and specificity; LFA, lateral flow assay; N protein, nucleocapsid protein.

Line 220: Please precise the number of NP/OP swabs and saliva samples tested.

Answer) We have incorporated this suggestion as follows: *"Clinical samples (COVID-19 patients (n = 42) and healthy controls (n = 20)), including NP/OP swabs (n = 30) and saliva samples (n = 12), were used to validate BEETLES²'s clinical applicability (Fig. 4)." (line 250–252).*

Line 234: Phrasing " We optimized the cut-off value for on-site COVID-19 diagnostics by maximizing the sum of sensitivity and specificity " is not clear. Revise

Answer) We have revised the expression to *"We acquired the cut-off value by maximizing the sum of sensitivity (true positive) and specificity (true negative) of COVID-19 tests." (line 264–265).*

Line 242: I do not agree with the following sentence "Assay using saliva samples has great advantages in easy sample collection". The collection of saliva is not always easier because you have to salivate a lot to produce the necessary volume. This is seen especially in symptomatic patients. The use of sponge solved this problem. It should be explained how the authors envisage the use of BEETLES following this type of sample (sponge).

Answer) We have revised the section to *"Second, we validated sensitivity using saliva samples (Fig. 4b). An assay using saliva samples has significant advantages for easy and non-invasive sample collection³⁹; however, salivary assays need additional sample collection steps such as prefiltration. For the prefiltration of cells and debris, we used a 450-nm filter membrane. Reportedly, tests using saliva samples have higher detection sensitivity in the first 10 days after infection⁴⁰. Compared to the commercial LFA's sensitivity before enrichment (3/12 = 25%), all the samples tested using BEETLES² had higher sensitivity (10/12 = 83.3%), indicating performance enhancement post sample enrichment. For the practical use of salivary assays with BEETLES², we suggested a simple saliva collection and purification system (Pure-SAL, Oasis Diagnostics) (Fig. S11a)" (line 272–280).*

[39] Soares Nunes, Lazaro Alessandro, Sayeeda Mussavira, and Omana Sukumaran Bindhu. "Clinical and diagnostic utility of saliva as a non-invasive diagnostic fluid: a systematic review." *Biochemia medica* 25.2 (2015): 177-192. DOI: <https://doi.org/10.11613/BM.2015.018>

Line 239: Please express the viral load in copies/mL instead of Ct.

Answer) We discussed the viral load with two Seoul St. Mary's Hospital clinicians. They mentioned that we could not quantify viral load in copies/mL since the viral RNA from the host cell can also be extracted. Therefore, we believe that Ct value is the best method for approximating virus titers in clinical samples (NP/OP and saliva).

Line 265: In the following sentence "Our system meets World Healthcare Organization (WHO)...", please precise the tem our system.

Answer) We changed to *"Our proposed BEETLES² system meets the World Healthcare Organization's (WHO) 'desirable criteria' guidelines for POCT"* (line 299–301).

Line 269: What is meant by "selectivity" here-sentence confusing?

Answer) We have deleted "selectivity" from the aforementioned sentence. Instead, we have used "specificity" and "cross-reactivity". We have revised the sentence to *"Another important parameter for COVID-19 diagnostics is cross-reactivity (Fig. 4j) and specificity (Fig. 4f)."* (line 304-305).

Line 275: Phrasing " By reducing false positives, " is not clear. Revise

Answer) We have revised the phrase from *"By reducing false positives, we increased the specificity."* to *"specificity is known as the true negative rate. Therefore, we increased the specificity by reducing false positives"* (line 309–310).

We also revised *"We observed no cross-reactivity and no selectivity problems from other viruses, demonstrating the high potential to meet the WHO criteria."* to *"We observed no cross-*

reactivity and specificity problems for other viruses, demonstrating a high potential to meet WHO criteria." (line 314–315).

Line 278: The following sentence “Chen et al. reported a quantitative and ultrasensitive in situ immunoassay using an AAO membrane, showing the ability to enrich SARS-CoV-2 viruses” should be placed in Line 77 when authors introduced AAO.

Answer) Based on the reviewer's suggestions, we have added previous works on AAO separation in the Introduction as *"Recently, Chen et al. reported a quantitative and ultrasensitive in situ immunoassay using a nanoporous anodic aluminum oxide (AAO) membrane, showing the ability to enrich SARS-CoV-2 viruses⁹. Generally, AAO membranes have been considered a promising material to facilitate size-based separation and enrichment^{24, 25, 26}."*(line 75–79).

Next, we have revised “With a combination of a high flux of AAO and aquaporin (AQP) water channel, BEETLES² allows fast water transport selectively within 3 min, achieving enrichment of SARS-CoV-2 viruses, nucleocapsid (N) proteins and, IgG antibodies.” To “*We prepared BEETLES² with a combination of red blood cell membranes (RBCMs) and nanoporous anodic aluminum oxide (AAO). The combination of high AAO flux and an aquaporin (AQP) water channel, unlike size-based AAO membrane enrichment^{27,28}, in BEETLES² allows fast water transport with additional permselectivity within 3 min, achieving enrichment of SARS-CoV-2 viruses, nucleocapsid (N) proteins, and immunoglobulin G (IgG) antibodies.*” (line 82–87).

Third, we have revised from “Chen et al. reported a quantitative and ultrasensitive in situ immunoassay using an AAO membrane, showing the ability to enrich SARS-CoV-2 viruses.⁹” to “*An AAO membrane has been used to develop an immunoassay for SARS-CoV-2 detection in saliva by enriching SARS-CoV-2 viruses.⁹ Following that study, we prepared only an AAO membrane and observed virus purification and enrichment via RT-qPCR (Fig.4k). In PBS buffers, the intact viruses could be enriched through the AAO membrane, as presented. However, we could not enrich virus particles using the AAO membrane in a lysate buffer (VTM) due to a lack of VTM lysates viruses for enrichment. Therefore, the Ct value using the AAO membrane did not change significantly. In many real clinical cases, samples are delivered with*

inactivated lysate (i.e., VTM). The Ct value was improved using BEETLES², even with VTM, up to 5.1, theoretically corresponding to 34.3 enrichment concentrations." (line 316–324).

Line 299: Please add a sentence describing the benefit of smartphone connection by describing more in depth REASSURED criteria, which are only mentioned in line 316.

Answer) We have revised to *"Moreover, the combination of AI-assisted smartphone applications⁴⁵ with BEETLES² can precisely predict on/off switching and perform quantitative classification, demonstrating great potential for the REASSURED (Real-time connectivity, Ease of specimen collection, Affordable, Sensitive, Specific, User friendly, Rapid, Equipment free and Delivered)⁴⁶, a new criterion for digital connectivity, and possibly detect asymptomatic transmission with simple frequent test strategies,^{8,42} which is difficult with RT-qPCR testing."* (line 356–366).

Line 313: The authors should explain how they figure an additional cost of tens of cents

Answer) We expect the additional cost for AAO materials and RBCM to reach tens of cents. However, as the review mentioned, this is only the material cost. We have revised the sentence from *"We expect it needs an additional cost of tens of cents onto commercial LFA, which meets the inexpensive criteria."* to *"We expect BEETLES² to be commercialized at a nominal additional cost, thereby meeting the inexpensive criteria."* (line 359–360)

Figure 1: Please clarify the abbreviation "BSA"

Answer) We have mentioned *bovine serum albumin (BSA) (~66.5 kDa)* in its first appearance. (line 111-113)

Figure 1C: The use of smartphones to improve on the one hand the reading of the results of rapid diagnostic tests and on the other hand, the transmission of the results to a laboratory computer system is a well-known subject. This point is not in my eyes the heart of the article

and brings little added value to the description of the Beetles. It would be wise to delete part C of figure 1. Nevertheless, discuss a possible connection of the quick test at the end of the chapter finding before the conclusion.

Answer) As reviewer suggested, we have deleted "smartphone" from manuscript line 100 and fig.1c.

In addition to the edition of these various points, the authors should also add a paragraph describing how they evaluated the potential leaks of liquids through the BEETLES device. This is extremely important because the notion of User-friendly taken up in the REASSURED criteria also includes the safety of the user, especially when the latter handles potentially infected samples such as respiratory samples or blood.

Answer) We completely agree with reviewer's comments. We have added a new paragraph at the end of Results as follows: *"To commercialize BEETLES², we should address the following issues: 1) pressure endurable design. Since the BEETLES² assay is performed under applied pressure, no liquid sample leakage can be tolerated. Especially when handling infectious samples, a leak-free design is needed for safety. 2) The volume issue in serological rapid diagnostic tests. For serological rapid diagnostic tests, 1–2 drops (<100 µL) a typically acquired from a finger prick; therefore, we need to develop devices for blood-based tests with small sample volumes. A microfluidic device integrated with the BEETLES² membrane is an ideal candidate for serological assays, 3) Shelf life is a key parameter to commercialize BEETLES². We obtained stability, for up to 15 days, under room temperature. However, the expected shelf life can be extended further with additional tests (Fig. S13)."* (line 337–346)

Conclusion:

With attention to the above details, the paper is worth to be published.

We want to thank you again for your valuable comments, and we found the comments very helpful in clarifying the originality of our work.

Reviewer #2

Paper 1: PCR-like Performance of Rapid Test with Permselective Tunable Nanotrap

Summary

The manuscript presents a nano trap with interesting permselectivity that can be used with commercial LFA. They use an RBCM lipid membrane on top of the AAO membrane to filter and enrich before using LFA. It reports bioengineered tools comprising nanoporous AAO membranes functionalized with red blood cell membranes to improve the antigen/antibody for the lateral flow assay with enhanced sensitivity and selectivity. The reported tools can eliminate the interfering proteins and increase the targets with a 20-fold enhancement of the detection limit. Clinical samples with different sample types and variants are tested using the proposed tool, validating its practical applications for a rapid diagnostics test. Overall, the authors claim to compete with PCR in terms of performance. The clinical claims of the manuscript are promising. However, some significant revision comments need to be addressed.

Answer) Thank you for your feedback on the manuscript. We appreciate your recognition of the potential of our work with permselective properties for use with commercial LFAs. We have considered your comments and revised the manuscript accordingly. We provide point-by-point responses, and all modifications in the manuscript have been highlighted in red.

Major Comments

1. The manuscript and the reported device are well characterized. I can predict that the reported device shows great potential for practical application for home diagnostics. However, a similar mechanism has been reported (Chen et al. Quantitative and ultrasensitive in situ immunoassay technology for SARS-CoV-2 detection in saliva. *Sci. Adv.* 8, eabn3481 (2022).) although the addition of red blood cell membranes can help to enrich the specific targets. Please show a competitive analysis concerning many similar works and justify why it has authentic merits.

Answer) Based on the reviewer's suggestions, we have added previous works on AAO separation in the Introduction as *"Recently, Chen et al. reported a quantitative and ultrasensitive in situ immunoassay using a nanoporous anodic aluminum oxide (AAO) membrane, showing the ability to enrich SARS-CoV-2 viruses⁹. Generally, AAO membranes have been considered a promising material to facilitate size-based separation and enrichment^{24, 25, 26}."*(line 75–79).

Next, we have revised “With a combination of a high flux of AAO and aquaporin (AQP) water channel, BEETLES² allows fast water transport selectively within 3 min, achieving enrichment of SARS-CoV-2 viruses, nucleocapsid (N) proteins and, IgG antibodies.” To “*We prepared BEETLES² with a combination of red blood cell membranes (RBCMs) and nanoporous anodic aluminum oxide (AAO). The combination of high AAO flux and an aquaporin (AQP) water channel, unlike size-based AAO membrane enrichment^{27,28}, in BEETLES² allows fast water transport with additional permselectivity within 3 min, achieving enrichment of SARS-CoV-2 viruses, nucleocapsid (N) proteins, and immunoglobulin G (IgG) antibodies.*” (line 82–87)

Third, we have revised from “Chen et al. reported a quantitative and ultrasensitive in situ immunoassay using an AAO membrane, showing the ability to enrich SARS-CoV-2 viruses.⁹” to “*An AAO membrane has been used to develop an immunoassay for SARS-CoV-2 detection in saliva by enriching SARS-CoV-2 viruses.⁹ Following that study, we prepared only an AAO membrane and observed virus purification and enrichment via RT-qPCR (Fig.4k). In PBS buffers, the intact viruses could be enriched through the AAO membrane, as presented. However, we could not enrich virus particles using the AAO membrane in a lysate buffer (VTM) due to a lack of VTM lysates viruses for enrichment. Therefore, the Ct value using the AAO membrane did not change significantly. In many real clinical cases, samples are delivered with inactivated lysate (i.e., VTM). The Ct value was improved using BEETLES², even with VTM, up to 5.1, theoretically corresponding to 34.3 enrichment concentrations.*” (line 316–324).

2. Page 6, Line 128, “The optimal RBCM concentration was checked by AFM ..” – It is mentioned that AFM is the deciding factor on optimal thickness. It is not clear how AFM observation can justify optimal thickness. In AFM images, even a 1 % sample seems pretty covered in AFM but not in SEM. Fig. 2d maybe not be necessary in that case. It would be helpful to mention the thickness with error %.

Answer) To avoid confusion, we have revised the section “We performed topological characterizations of BEETLES² membrane with different RBCM concentrations (0-4% (v/v)), showing fully coated RBCM on AAO materials over 2% concentration (Fig. 2d and Fig. S3). The optimal RBCM concentration was checked by AFM-based surface roughness analysis depending on the RBCM concentration (Fig. S4). In addition, using Kelvin probe force microscopy (KPFM), we measured physicochemical characterizations of BEETLES² membrane (Fig. 2e-f and Fig. S5), showing that it became negatively charged because of the

negatively charged phospholipids in RBCM.²³" to "From the SEM images (Fig. S4), we observed conformal coating of RBCM with the least defects over 2% RBCM concentration. 4% RBCM also showed conformal coating; however, it slowed water transport compared to 2% RBCM. For additional topological analysis, we measured the surface roughness of the BEETLES² membrane depending on the RBCM concentration (0 – 4% (v/v)) via AFM (Fig. 2d, S5 and 6). Over the scanned $10 \times 10 \mu\text{m}$ area, the surface root means square roughness (R_q) of each BEETLES² membrane significantly decreased compared to that of a bare AAO membrane, indicating more conformal RBCM coating on the surface of AAO membrane. Along with AFM and SEM analysis, we conducted Kelvin probe force microscopy (KPFM) (Fig. 2e–f and S7). Via KPFM, we measured the physicochemical characterizations of the BEETLES² membrane, which was negatively charged due to the negatively charged phospholipids in RBCM^{29,30}. In contrast, the BEETLES² membrane with 4% RBCM showed a relatively positively charged surface compared to that with 2% RBCM. The flow of current between the KPFM cantilever tip and sample is speculated to have slowed down because the nonconductive RBCM was excessively wrapped in a dried state on the surface of the AAO membrane³¹. We selected 2% RBCM as the optimal condition because a negatively charged surface potential of the BEETLES² membrane is significant for maximizing the electrostatic interaction with the positively charged nucleocapsid protein." (line 132–149)

3. Fig. 3f, Why do KPFM measurements show a reduced magnitude of negative charge for the 4% RBCM sample? How significant is the importance of potential for achieving the desired permselectivity?

Answer) The results show that the surface of the BEETLES² membrane becomes negatively charged (0 – 2% (v/v)) since a major component of RBCM is negatively charged phospholipids [1, 2]. In contrast, the 4% BEETLES² membrane shows relatively positive surface charge. Additionally, the flow of current between the KPFM cantilever tip and sample is speculated to have slowed down because the nonconductive RBCM was excessively wrapped in a dried state on the surface of the AAO membrane [3].

The negatively charged surface potential of the BEETLES² membrane is significant for maximizing the electrostatic interaction with positively charged nucleocapsid proteins.

- [1] Jang, Jae Won, et al. "Surface Functionalization of Enzyme-Coronated Gold Nanoparticles with an Erythrocyte Membrane for Highly Selective Glucose Assays." *Analytical Chemistry* 94.17 (2022): 6473-6481. DOI: <https://doi.org/10.1021/acs.analchem.1c04541>
- [2] Fernandes, Heloise Pöckel, Carlos Lenz Cesar, and Maria de Lourdes Barjas-Castro. "Electrical properties of the red blood cell membrane and immunohematological investigation." *Revista brasileira de hematologia e hemoterapia* 33 (2011): 297-301. DOI: <https://doi.org/10.5581/1516-8484.20110080>
- [3] Kalinin, Sergei V., and Nina Balke. "Local electrochemical functionality in energy storage materials and devices by scanning probe microscopies: status and perspectives." *Advanced Materials* 22.35 (2010): E193-E209. DOI: <https://doi.org/10.1002/adma.201001190>

4. There seems to be a drastic change in RBCM deposition from 1 % to 2% compared to other consecutive pairs (SEM images, Fig. S3). The samples of 0.5 % and 1 % seem very similar. Please justify the reason. If 2% is the first sample to be fully covered, it would be ideal to choose a % slightly above that as optimum.

Answer) We have added additional SEM images for 1%, 1.5%, and 2% RBCM concentrations. From the SEM images, we observed conformal coating of RBCM with the least defects for 2% RBCM concentration. We agreed with reviewer's observation of "2% is the first sample to be fully covered". As mentioned, we added sentences as *"From the SEM images (Fig. S4), we observed conformal coating of RBCM with the least defects over 2% RBCM concentration. 4% RBCM also showed conformal coating; however, it slowed water transport than 2% RBCM."* (line 132–134)

Supplementary Fig. 4. SEM images of the BEETLES² membrane surface for optimizing RBCM concentration. (a) 1% RBCM coating, (b) 1.5% RBCM coating, and (c) 2% RBCM coating. SEM, scanning electron microscope; BEETLES², bioengineered enrichment tools for the LFA with enhanced sensitivity and specificity; RBCM, red blood cell membrane.

5. Elaborate on the novel functionality that the RBCM membrane adds to the filter. If the 20 nm AAO membrane is used as-is, size-based filtration is possible. But that might cause nucleocapsid protein to escape if there are lots of them in the saliva sample. It will be helpful to describe how many direct nucleocapsid proteins can be there in the saliva sample to justify the RBCM layer.

Answer) First, throughout the manuscript, we have highlighted BEETLES²'s powerful permselectivity and tunability by adding the RBCM membrane, resulting in a sensitive binary response to commercial LFA. In fig. 3, we demonstrated that positively charged N proteins (Fig. 3f) were enriched 24-fold, measured by nanodrops regardless of applied pressure (0.1–3 bar). With a similar molecular weight, enrichment with only N proteins was up to 24.16-fold, while BSA showed no significant enrichment. These results indicate that the BEETLES² membranes completely blocks the penetration of positively charged proteins through the filters from the electrostatic interaction of negatively charged phospholipids and positively charged viral proteins under physiological conditions³². The enriched samples from N proteins were acquired 3 min after BEETLES² operation (SDS-PAGE in Fig. 3e). In contrast, no enrichment was observed in BSA over the enrichment zone after 3 min. (Fig. 3).

Second, in response to the concern of "But that might cause nucleocapsid protein to escape if there are lots of them in the saliva sample.", we believe that a significant amount of N proteins might escape by avoiding the electrostatic interaction in BEETLES². While we did not demonstrate a significant amount of N proteins, as shown in fig.3j and fig.4b, in the salivary assay, we observed significant advances even with clinical samples. Furthermore, in the rapid test, a higher concentration of N proteins causes the Hook effect; therefore, we believe that a higher concentration cannot affect the sensitivity in binary tests.

6. Small syringes can have a much larger pressure than 3 bar; even Fig. S8 shows a 6-bar pressure. Fig. 3b shows that increasing pressure can tune the enrichment. It helps to remove BSA, but it can cause NP to de-enrich. There is a trend of de-enrichment of NP in Fig. 3f as pressure increases. What is the possibility that increased pressure might allow nucleocapsid protein to pass through, overriding the electrostatic interaction between NP and RBCM? The system may be very pressure sensitive in normal pressure ranges. It is important to study that aspect.

Answer) We apologize for this error. To correct the same, we have revised actual manual pressure as *"We calculated the average hand-powered pressure from five individuals (4 men and 1 woman) with BEETLES² as 2.3 ± 0.2 bar (See revised Fig. S14)."* (line 473–474).

For rapid testing, the sample enrichment time is also important. A higher pressure results in faster enrichment time. We have optimized the pressure to hand-actuable levels while maintaining stability and preventing leakage in the BEETLES² device. Further, Fig. R1 (reviewer only) clearly illustrates the effect of higher pressure on nucleocapsid proteins. We applied pressure up to 4.5 bar since the maximum operating pressure of the stirred cell was 4.4 bar. As expected, we observed no significant decrease in enrichment even at 4.5 bar.

Lastly, we have added the following: *"We showed that the permselectivity of BSA and N proteins was maintained under hand-powered pressure."* (line 474–475)

Supplementary Fig. 14. Average hand-powered pressure of 2.3 ± 0.2 bar for 3 min using the BEETLES² device from five individuals (4 men and 1 woman). BEETLES², bioengineered enrichment tools for the LFA with enhanced sensitivity and specificity.

Fig. R1 N protein enrichment test with high pressure of 3–4.5 bar

7. Fig. 3b, 3d, 3f, 3g- Is the effect of pressure on enrichment far more overriding than other aspects- charge and pH? It is necessary to increase the pressure window and determine if charge and pH are still making a significant difference.

Answer) The parameters of higher significance are size and charge, not pressure. We observed only tunable permselectivity of negatively charged BSA with applied pressure. For BSA samples, we filtered out BSA with a pressure >0.5 bar. The most probable explanation is that cell squeezing enables the delivery of various materials, representing the need for applying pressure to enhance negatively charged protein permeation. With increasing pressure, BSA freely passed through the BEETLES² membrane without significant enrichment. As mentioned in the response to query 6, we have verified the permselectivity of N protein with higher pressure up to 4.5 bar.

8. Page 2, line 34- The tunability of nanotrap is one of the claims. Enrichment tunability under different pressures is a standard principle used in filtration. There is no new insight added to that tunability in the manuscript. Hence can be avoided as a claim.

Answer) Thank you for this input. As mentioned, enrichment tunability under different pressures is a standard principle used in filtration. With increasing pressure, we can increase the rate of separation. However, our intended meaning was tunability between enrichment and

filter-out, not enrichment tunability under different pressures. We attempted to find the accurate expression for fig. 3b data with fig. 2b illustration and determined the use of "tunable" and "tunability" from two previous papers [1–2]. Hence, we would like to use "tunable" and "tunability" if acceptable by the reviewer.

[1] Tuneable Elastomeric Nanochannels for Nanofluidic Manipulation, *Nat. Mater.* 2007

[2] Tunable Negative Permittivity in Flexible Graphene/PDMS Metacomposites, *J. Phys. Chem. C* 2019, 123, 38, 23635–23642

9. Page 16, line 421- While transferring the enriched sample to LFA, the electrostatic force should still make the NPs attached to the RBCM layer, although it is inversed. How is that overridden? It should be explained. How much pressure is applied while LFA application?

Answer) We have revised manuscript from "We first connected BEETLES² membrane to the sample reservoir and pushed the plunger for enrichment. After 3 min operation, we connected BEETLES² membrane reversely onto the running buffer reservoir." to "*We first connected the BEETLES² membrane to the sample reservoir and pushed the plunger for enrichment. After 3 min of operation, we reversed the direction of BEETLES² membrane attachment to the running buffer reservoir. Since the commercial running buffer included a surfactant, we could successfully demonstrate the N protein assay regardless of the electrostatic force between the N protein and RBCM layer*" (line 477–481).

To precisely control the pressure, we applied a constant pressure of ~3 bar via a regulator throughout the experimentation in this study. Next, we demonstrated the feasibility of "Hand-powered portable syringes" as a low-cost alternative for point-of-care diagnostics. Accordingly, we applied an average pressure of ~2.3 bar, as shown in *Fig. S14*. With further investigation, we aim to develop a simplified prototype operated with a pressure-driven force by a vacuum chamber.

10. Explain why Fig. 4f shows no cross-reactivity.

Answer) For Fig. 4f, we used "specificity" and not "cross-reactivity". As elaborated in line 305, To check specificity, we tested the colorimetric signals from healthy controls (n = 20) showing 100% specificity (true negatives = 20/20) (Fig. 4f). By reducing false positives, we increased the specificity.

11. How is the long RBCM layer stable at room temperatures? A study about the storage of the final membrane outside can supplement the analysis.

Answer) As the reviewer highlighted, shelf life is a key parameter to commercialization. We have added new paragraph at the end of Results with newly added *Supplementary Fig. 13* as follows: *"To commercialize BEETLES², we should address the following issues: 1) pressure enduring design. Since the BEETLES² assay is performed under applied pressure, no liquid sample leakage can be tolerated. Especially when handling infectious samples, a leak-free design is needed for safety. 2) The volume issue in serological rapid diagnostic tests. For serological rapid diagnostic tests, 1–2 drops (<100 μ L) a typically acquired from a finger prick; therefore, we need to develop devices for blood-based tests with small sample volumes. A microfluidic device integrated with the BEETLES² membrane is an ideal candidate for serological assays, 3) Shelf life is a key parameter to commercialize BEETLES². We obtained stability, for up to 15 days, under room temperature. However, the expected shelf life can be extended further with additional tests (Fig. S13)."* (line 337–346)

Supplementary Fig. 13. Shelf-life test for commercializing the BEETLES² membrane. (a) N protein enrichment performance test of the BEETLES² membrane incubated at room temperature. (b) SEM images. BEETLES², bioengineered enrichment tools for the LFA with enhanced sensitivity and specificity; SEM, scanning electron microscope; N protein, nucleocapsid protein.

Minor Comments

1. Fig. 2e and 2f convey similar information, and one can be avoided.

Answer) Thank you for your comment. We would like to highlight that Fig. 2e shows the uniform distribution of surface charge when the AAO surface is coated with RBCM. Meanwhile, Fig. 2f shows the quantitative analysis of surface charge depending on RBCM concentration (0–4%). To enhance the readability of our manuscript, we decided to incorporate both figures.

2. Please number supplementary sub-figures.

Answer) We have corrected it.

3. Please increase the size of the text inside the figures.

Answer) We have corrected it.

4. Fig. 2g and 2h convey similar information, and one can be removed.

Answer) To illustrate good performance of FRAP imaging despite the fluidity of the cell membrane, the graph and fluorescence image were added together. Hence, if the reviewer accepts, we would like to depict *Fig. 2g* as below.

Conclusion

The manuscript presents exciting ideas. However, the enrichment, tenability, selectivity, and specificity observations should be explained more scientifically. The choice of RBCM concentrations should be defined more convincingly. The pressure range that they tested the system needs to be evaluated. After a critical major revision, the manuscript can be considered.

Answer) We really appreciate the reviewer's valuable comments. The comments were very helpful in enhancing our manuscript. We hope the manuscript, with careful revisions, meets your high standards.

Reviewer #3

This paper reports the development of an LFA tool for sensitive and selective COVID-19 detection. This method uses a perm-selective tunable Nanotrap to enhance clinical testing performance for infectious agents. The results described purport the enablement of more accurate and rapid point-of-care testing, and for a lower cost than RT-qPCR. The narrative should not convey that the BEETLE system allows for results comparable to RT-qPCR, as LFA (antigen) tests will always have drawbacks associated with a passive flow microfluidics, sensitivity and visual detection, however, the ability to rapidly enrich samples using the perm-selective Nanotrap system certainly aids sensitivity and could enhance clinical performance.

Overall, the paper is well written, although it is not conversational – the narrative unclear and somewhat ‘dry’ to this reviewer – but this could simply be stylistic differences. The figures are detailed and easily digestible, however, resolution could be improved. The supplemental video was very helpful.

Answer) Thank you for your feedback on our manuscript. We appreciate your recognition of the potential of our LFA tool for sensitive and selective COVID-19 detection via a permselective tunable Nanotrap to enhance clinical testing performance. We provide point-by-point responses, and all modifications in the manuscript have been highlighted in red.

Major Issues:

The intro provides a good overview of the motivation for the project, but it does not introduce much about the method selected for COVID-19 enrichment. For a more complete introduction, it should provide some background description of the tunable Nanotrap (prior use, ‘mechanism’ (how it works, etc.). Additionally, while it is appropriate to discuss the ‘WHO criteria’ for POCT (actually, multiple times at the end of the paper), it would be beneficial to discuss these in the introduction as well.

Answer) Based on the reviewer's suggestions, first, we have added the “WHO criteria” in the Introduction as *"The LFA platforms are considered the best candidate as they meet the WHO (World Health Organization) "ASSURED" criteria (affordable, sensitive, specific, user-friendly, rapid and robust, equipment-free, and deliverable to end-users)." (line 67–70).*

Further, considering comments from all the reviewers, we have revised the Introduction by adding recent journal papers on COVID-19 enrichments technique. We have added previous

studies on AAO separation as follows: *"Recently, Chen et al. reported a quantitative and ultrasensitive in situ immunoassay using a nanoporous anodic aluminum oxide (AAO) membrane, showing the ability to enrich SARS-CoV-2 viruses⁹. Generally, AAO membranes have been considered a promising material to facilitate size-based separation and enrichment^{24, 25, 26}."* (line 75–79).

Second, we have revised *"With a combination of a high flux of AAO and aquaporin (AQP) water channel, BEETLES² allows fast water transport selectively within 3 min, achieving enrichment of SARS-CoV-2 viruses, nucleocapsid (N) proteins and, IgG antibodies."* To *"We prepared BEETLES² with a combination of red blood cell membranes (RBCMs) and nanoporous anodic aluminum oxide (AAO). The combination of high AAO flux and an aquaporin (AQP) water channel, unlike size-based AAO membrane enrichment^{27,28}, in BEETLES² allows fast water transport with additional permselectivity within 3 min, achieving enrichment of SARS-CoV-2 viruses, nucleocapsid (N) proteins, and immunoglobulin G (IgG) antibodies"* (line 82–87).

Third, we have revised *"Chen et al. reported a quantitative and ultrasensitive in situ immunoassay using an AAO membrane, showing the ability to enrich SARS-CoV-2 viruses.⁹"* to *"An AAO membrane has been used to develop an immunoassay for SARS-CoV-2 detection in saliva by enriching SARS-CoV-2 viruses.⁹ Following that study, we prepared only an AAO membrane and observed virus purification and enrichment via RT-qPCR (Fig.4k). In PBS buffers, the intact viruses could be enriched through the AAO membrane, as presented. However, we could not enrich virus particles using the AAO membrane in a lysate buffer (VTM) due to a lack of VTM lysates viruses for enrichment. Therefore, the Ct value using the AAO membrane did not change significantly. In many real clinical cases, samples are delivered with inactivated lysate (i.e., VTM). The Ct value was improved using BEETLES², even with VTM, up to 5.1, theoretically corresponding to 34.3 enrichment concentrations. "* (line 316–324).

With the main findings, it's not clear exactly how the enriched fractions of samples are being further processed for analysis and subsequent detection. While this is clarified somewhat by the video, the body of the text should describe the unit operations associated with handheld device. This would nicely compliment moving supplemental figure 7 to the main body of text for ease of visualization.

Answer) To clearly describe the explanation and process, we have revised the manuscript "We first connected BEETLES² membrane to the sample reservoir and pushed the plunger for enrichment. After 3 min operation, we connected BEETLES² membrane reversely onto the running buffer reservoir." to *"We first connected the BEETLES² membrane to the sample reservoir and pushed the plunger for enrichment. After 3 min of operation, we reversed the direction of BEETLES² membrane attachment to the running buffer reservoir. Since the commercial running buffer included a surfactant, we could successfully demonstrate the N protein assay regardless of the electrostatic force between the N protein and RBCM layer"* (line 477–481). While we cannot conclude the exact fraction of enriched targets, we expect most of the enriched targets to be extracted since the assay was greatly enhanced proportional to the preconcentration factor.

With regards to "This would nicely compliment moving supplemental figure 7 to the main body of text for ease of visualization.", we would like to retain Fig. S7 (Fig. S11 now) if acceptable by the reviewer. Throughout the manuscript, we have applied a constant pressure of ~3 bar via a regulator and stirred the cell to precisely control the pressure. Then, we showed the feasibility of "Hand-powered portable syringes" for low-cost alternatives for point-of-care diagnostics.

Further, we have added new paragraph at the end of Results as *"To commercialize BEETLES², we should address the following issues: 1) pressure endurable design. Since the BEETLES² assay is performed under applied pressure, no liquid sample leakage can be tolerated. Especially when handling infectious samples, a leak-free design is needed for safety. 2) The volume issue in serological rapid diagnostic tests. For serological rapid diagnostic tests, 1–2 drops (<100 µL) a typically acquired from a finger prick; therefore, we need to develop devices for blood-based tests with small sample volumes. A microfluidic device integrated with the BEETLES² membrane is an ideal candidate for serological assays, 3) Shelf life is a key parameter to commercialize BEETLES². We obtained stability, for up to 15 days, under room temperature. However, the expected shelf life can be extended further with additional tests (Fig. S13)."* (line 337–346)

There is a lack of clarity as to which buffer is used and what the context is for use of the different range of buffers used in these types of applications and how this aids in assay flexibility. [Page 5 (lines 101 – 102)]

Answer) In Fig. 4k, we have illustrated the flexibility of the proposed BEETLES² compared with bare AAO (Science Advances 8, eabn3481, 2022). To summarize, AAO can only implement size separation; therefore, it can only enrich the virus, not the N proteins in VTM (lysate buffer). Meanwhile, BEETLES² can enrich N proteins in VTM buffer, indicating assay flexibility. To clarify this point, we have the sentence as *"An AAO membrane has been used to develop an immunoassay for SARS-CoV-2 detection in saliva by enriching SARS-CoV-2 viruses.⁹ Following that study, we prepared only an AAO membrane and observed virus purification and enrichment via RT-qPCR (Fig.4k). In PBS buffers, the intact viruses could be enriched through the AAO membrane, as presented. However, we could not enrich virus particles using the AAO membrane in a lysate buffer (VTM) due to a lack of VTM lysates viruses for enrichment. Therefore, the Ct value using the AAO membrane did not change significantly. In many real clinical cases, samples are delivered with inactivated lysate (i.e., VTM). The Ct value was improved using BEETLES², even with VTM, up to 5.1, theoretically corresponding to 34.3 enrichment concentrations."* (line 316–324).

Also, we have added the following in the revised manuscript: *"The primary advantage of BEETLES² is that the viral protein and virus itself can be enriched under various buffers including phosphate buffered saline (PBS), saliva, serum, and viral transport medium (VTM)."* (line 107–109).

The definition of “tunable” is not clear from the descriptions provided. The authors need to include more text describing exactly what is being ‘tuned’ and how that can be applied to different analytes. [Page 6 (lines 115 – 117)]

Answer) We thank you for your thoughtful comment. Our intended meaning was tunability between enrichment and filter-out, not enrichment tunability under different pressures. We attempted to find the accurate expression for fig. 3b data with fig. 2b illustration and determined the use of **"tunable"** and **"tunability"** from two previous papers [1–2]. Hence, we would like to use **"tunable"** and **"tunability"** if acceptable by the reviewer.

[1]Tuneable Elastomeric Nanochannels for Nanofluidic Manipulation,” Nat. Mater. 2007

[2] Tunable Negative Permittivity in Flexible Graphene/PDMS Metacomposites, J. Phys. Chem. C 2019, 123, 38, 23635–23642

To clarify, we have added the following in the revised manuscript: *"Unlike the negatively-charged BSA’s tunable properties between enrichment and separation with pressure, positively-charged N proteins showed only enrichment without tunability."* (line 208–210).

From the text, it appears that several studies were conducted to determine the optimal RBCM concentration, and it seems the 2% concentration was based on KPFM and fluorescent data alone. If this is the case, either provide an explanation to describe why all other data was excluded, or link this data back to the other experiments (Figs 2d, s3, s4, 2e – f, and s5, s6). [Page 6 (lines 136 – 137)]

Answer) We have revised the manuscript as follows: *"From the SEM images (Fig. S4), we observed conformal coating of RBCM with the least defects over 2% RBCM concentration. 4% RBCM also showed conformal coating; however, it slowed water transport than 2% RBCM."* (line 132–134). We added additional SEM images for 1%, 1.5%, and 2% RBCM concentration. From those images, we concluded conformal coating of RBCM with the least defect over 2% RBCM concentration.

We also revised *"Based on the KPFM and fluorescent data, a 2% RBCM concentration was determined as best for BEETLES² membrane."* to *"Based on topological analysis, KPFM, and fluorescent data, 2% RBCM concentration was determined to be the best for the BEETLES² membrane."* (line 153–154).

Figure 3 – When testing patient samples, was each sample split into 3 aliquots to run with and without BEETLES²? While I recognize that the spread of data points is due to patient-dependent differences in viral load, stochastic differences between aliquots of this patient sample would be noteworthy data points. If this was not done, clarify that this is an n = 1 for each patient sample.

Answer) First, we showed the run-to-run error (n = 3) from the same samples (split into 3 aliquots) in Fig. 3, as follows: *"The error bar in Fig. 3 represents the run-to-run deviation (n=3)"* (line 175–176).

Second, we have added the following in the revised manuscript: *"For the COVID-19 Ab test (Fig. 3i), we prepared seven concentrations of the COVID-19 antibody (sodium citrate plasma, Trina Bioreactives AG, Switzerland) sample with 1xPBS. We added 3 mL of each sample into a polycarbonate stirred cell (341000, STERLITECH, USA) mounted with an RBCM-coated AAO membrane and operated it for 3 min. Next, the enriched sample was recovered via a pipette with 100 µL LFA extraction buffer. Subsequently, we used the enriched sample in the*

COVID-19 Ab LFA kit (AllCheck COVID-19 IgG/IgM, Calth Inc., Korea) and noted the result at 10 min.

Similarly, for the salivary COVID-19 Ag test (Fig. 3j), we diluted seven concentrations of the COVID-19 N protein sample with artificial saliva (A7990, Solarbio, China). First, we added 3 mL of each sample into a polycarbonate stirred cell mounted with an RBCM-coated AAO membrane and operated it for 3 min. Next, the enriched sample was recovered via a pipette with 100 μ L of LFA extraction buffer. Subsequently, we used the enriched sample in the COVID-19 Ag LFA kit and noted the result at 10 min." (line 434–446).

Third, we performed a clinical test for each patient once. We have mentioned $n = 1$ in the revised Fig. 4 caption as *"We performed the clinical test for each patient once ($n = 1$)." (line 660–661)*

The verbiage associated with Figure 4e states that with the BEETLES² system, $\frac{3}{4}$ of true positives were called, whereas the commercial assay called 0/4. However, it appears from the figure that each condition has one sample residing on the threshold line, and it was called in favor or the manuscript narrative. This needs to be addressed.

Answer) We observed a clear positive colorimetric signal with BEETLES² (right: w/ BEETLES²; left: w/o BEETLES²). Note that both images are from a data point slightly below/above the cut-off line. We have revised the manuscript as: *"We elucidated data points near the cut-off line with LFA test results (Fig. S12)." (line 292–293).*

Supplementary Fig. 12. (a) Color intensities, (b) image w/o BEETLES² data slightly below the cut-off line, and (c) image w/ BEETLES² data slightly above the cut-off line. BEETLES², bioengineered enrichment tools for the LFA with enhanced sensitivity and specificity.

Please comment on whether the observation of “no cross-reactivity” was a result of the specificity of the downstream LFA test, or the result of permselectivity of the enrichment process described here. [Page 10 (lines 269 – 273)]

Answer) We have added the following in the revised manuscript: *"Both the LFA test and BEETLES² enrichment can cause cross-reactivity and specificity. However, since the LFA manufacturer strictly controlled cross-reactivity and specificity (>99%), we expect that the results of cross-reactivity and specificity are primarily due to BEETLES² enrichment. (Table. S1)"* (line 310–314)

Supplementary Table. 1. Performance characteristics of the commercial COVID-19 Ag LFA kit (Sensitivity: 97.5 %; Specificity: 100 %). Ag, antigen; LFA, lateral flow assay.

Clinical performance Result		RT-PCR method		Total
		Positive	Negative	
ALLCheck COVID-19 Ag	Positive	78	0	78
	Negative	2	60	62
Total		80	60	140

With respect to the ‘cost per test’ as described here, the authors need to provide details as to how they arrived at those numbers (e.g., would this be at low-rate production? high rate production)? and how easily these membranes could be manufactured.

Answer) We expect the additional cost for AAO materials and RBCM to reach tens of cents. However, as the review mentioned, this is only the material cost. We have revised the sentence from *"We expect it needs an additional cost of tens of cents onto commercial LFA, which meets the inexpensive criteria."* to *"We expect BEETLES² to be commercialized at a nominal additional cost, thereby meeting the inexpensive criteria."* (line 359–360)

Minor issues:

- The abbreviation “NP” is used to describe both Nasopharyngeal swabs and N proteins – distinguish.

Answer) As suggested, we have distinguished the abbreviation in the revised manuscript. We changed NP as N protein.

- Page 3 (Lines 45 – 47) Phrasing needs adjusted as the point is not clearly stated as is.

Answer) We have rephrased the sentence as follows: *“For commercial rapid testing of COVID-19, 59 antigen diagnostic tests for SARS-CoV-2 are available under Emergency Use Authorization (EUA) (as of 1 Jan 2022)”* (line 45–46).

- Figure 1B & 1C are repetitive aside from Ag vs. Ab test – either combine or remove 1C

Answer) We have deleted Fig. 1c in the revised manuscript.

- Page 5 (lines 102 – 108) We understand that the pore sizes are tunable based upon pressure; however, it would be advantageous to have some size range described in nm

Answer) Thank you for your input. A previous study [1] reported that the cell membrane could allow biomolecules such as proteins, nucleoids, and polysaccharides up to 2,000 kDa to pass through under squeezing pressure. To the best of our knowledge, physical breaches, including pressure and shear stress, induce nanoruptures of 1–10 nm [2]. Further research is required to investigate the pore-size range of RBCMs under pressure of ~3 bar.

[1] Joo, Byeongju, et al. "Highly efficient transfection of human primary T lymphocytes using droplet-enabled mechanoporation." *ACS nano* 15.8 (2021): 12888-12898.

[2] Ammendolia, Dustin A., William M. Bement, and John H. Brumell. "Plasma membrane integrity: implications for health and disease." *BMC biology* 19.1 (2021): 1-29.

- Page 9 (line 220) – Create an abbreviation for NP/OP swabs prior to use

Answer) We have mentioned the full form for NP/OP at its first instance.

- Page 9 (lines 229 – 232) – These sentences describing IRB approval and collection/protocols should be moved to the methods section

Answer) As suggested, we have moved the sentences regarding IRB approval in the methods section. (line 457–459)

- Page 9 (lines 232 – 233) – Make clear that you are naming your results here by using either quotation marks and/or italicized names

Answer) As suggested, we have italicized the naming of the results.

- Figure 3a – f – add a sentence to the body of the text to describe why there is such a large spread of LFA results (e.g., sample – to – sample variation from patient viral load)

Answer) We have added the following sentence in the manuscript: *"The error bar in Fig. 3 represents the run-to-run deviation (n=3)."* (line 175–176).

- Page 10 (line 253) typo – “that” should be “than”

Answer) We have corrected the error.

- Figure 4k – please add fluorescence units to the y axis of the qPCR plot and a threshold line for better visualization of Ct value differences

Answer) We have corrected the figure as follows:

- Figure 4l – Figure very difficult to interpret – consider either splitting this into two bar graphs or explaining how to read this in the figure legend

Answer) We corrected the figure as follows:

- Page 12 (line 309) – This conclusion alleges that blood serum samples were tested, but they were not in the body of work described here. Please remove.

Answer) We have used a pooled clinical sample in Fig. 3i (COVID-19 Ab test). We have added information regarding the sample in Materials and Methods as *"For the COVID-19 Ab test (Fig. 3i), we prepared seven concentrations of the COVID-19 antibody (sodium citrate plasma, Trina Bioreactives AG, Switzerland) sample with 1xPBS. We added 3 mL of each sample into a polycarbonate stirred cell (341000, STERLITECH, USA) mounted with an RBCM-coated AAO membrane and operated it for 3 min. Next, the enriched sample was recovered via a pipette with 100 µL LFA extraction buffer. Subsequently, we used the enriched sample in the COVID-19 Ab LFA kit (AllCheck COVID-19 IgG/IgM, Calth Inc., Korea) and noted the result at 10 min.*

Similarly, for the salivary COVID-19 Ag test (Fig. 3j), we diluted seven concentrations of the COVID-19 N protein sample with artificial saliva (A7990, Solarbio, China). First, we added 3 mL of each sample into a polycarbonate stirred cell mounted with an RBCM-coated AAO membrane and operated it for 3 min. Next, the enriched sample was recovered via a pipette with 100 µL of LFA extraction buffer. Subsequently, we used the enriched sample in the COVID-19 Ag LFA kit and noted the result at 10 min." (line 434–446)

Recommendation

This is interesting tech that could be beneficial with scenarios where lost-cost but rapid results are required. I believe this ms could clear the bar for NATCOMMS and, in this reviewer's opinion, resides somewhere between a 'major' and 'minor' revision. But that requires that the issues identified above are adequately addressed.

We appreciate the reviewer's valuable comments. They have been invaluable in enhancing the quality of our manuscript.

Reviewer #4

The manuscript by Park et al. has developed bioengineered enrichment tools for LFA with enhanced sensitivity and selectivity for rapid detection. The manuscript is well written and can be considered for publication following major revision.

Answer) Thank you for your feedback on our manuscript. We have revised the manuscript based on your feedback and hope that the revised version will meet your expectations.

Major concerns:

1. In the KPFM study, with increase in the concentration of RBCM coating the surface potential increased where 2% showed highest followed by a sudden drop on using 4% coating. What could be possible reason for such a sudden drop? Will there be an additive effect if the authors study any percentage between 2-4% coating?

Answer) The results showed that the surface of the BEETLES² membrane became negatively charged (0–2% (v/v)) because the major components of RBCM are negatively charged phospholipids [1, 2]. In contrast, the 4% BEETLES² membrane surface showed a relative positive charge. We speculate that the flow of current between the KPFM cantilever tip and sample slowed down because the nonconductive RBCM was excessively wrapped in a dried state on the AAO membrane surface [3].

The negatively charged surface potential of the BEETLES² membrane is significant for maximizing the electrostatic interaction with positively charged nucleocapsid proteins.

[1] Jang, Jae Won, et al. "Surface Functionalization of Enzyme-Coronated Gold Nanoparticles with an Erythrocyte Membrane for Highly Selective Glucose Assays." *Analytical Chemistry* 94.17 (2022): 6473-6481. DOI: <https://doi.org/10.1021/acs.analchem.1c04541>

[2] Fernandes, Heloise Pöckel, Carlos Lenz Cesar, and Maria de Lourdes Barjas-Castro. "Electrical properties of the red blood cell membrane and immunohematological investigation." *Revista brasileira de hematologia e hemoterapia* 33 (2011): 297-301. DOI: <https://doi.org/10.5581/1516-8484.20110080>

[3] Kalinin, Sergei V., and Nina Balke. "Local electrochemical functionality in energy storage materials and devices by scanning probe microscopies: status and perspectives." *Advanced Materials* 22.35 (2010): E193-E209. DOI: <https://doi.org/10.1002/adma.201001190>

2. For improving permeation, highest tested pressure of about 3 bar was used for the later clinical studies. However, it was observed from Figure 3 b, d, and f, pressure of about 0.5 bar

had shown to permeate BSA and able to enrich NP and IgG as like 3 bar. Why have the authors applied 3 bar pressure for later studies?

Answer) In original manuscript, we made mistake to calculate the hand-powered pressure by considering the instant hand-powered pressure. We have revised actual manual pressure as "*We calculated the average hand-powered pressure from five individuals (4 men and 1 woman) with BEETLES² as 2.3 ± 0.2 bar (See revised Fig. S14).*" (line 473–474). For rapid testing, the sample enrichment time is also important. A higher pressure results in faster enrichment time. We have optimized the pressure to hand-actuable levels while maintaining stability and preventing leakage in the BEETLES² device.

Supplementary Fig. 14. Average hand-powered pressure of 2.3 ± 0.2 bar for 3 min using the BEETLES² device from five individuals (4 men and 1 woman). BEETLES², bioengineered enrichment tools for the LFA with enhanced sensitivity and specificity.

3. In optimization highest selected pressure of 3 bar has been selected for the further studies, what will be the effect of pressure above 3 bar.

Answer) Fig. R1 (reviewer only) clearly illustrates the effect of higher pressure on nucleocapsid proteins. We applied pressure up to 4.5 bar since the maximum operating pressure of the stirred cell was 4.4 bar. As expected, we observed no significant decrease in enrichment

even at 4.5 bar. We have added the following: *"We showed that the permselectivity of BSA and N proteins was maintained under hand-powered pressure."* (line 474–475)

Fig. R1 N protein enrichment test with high pressure of 3–4.5 bar

4. AFM topography images, the images could be taken at higher magnification which can give a better idea of changing topography of AAO on RBCM coating.

Answer)

To provide better insight into AFM topography of the AAO membrane with different RBCM concentrations (0–4% (v/v)), we magnified the AFM topography images ($1 \mu\text{m} \times 1 \mu\text{m}$, 7.8 nm/pixel), as shown in Fig. S6. From the magnified images, we confirmed that the structure of the AAO membranes was partially observed at 0–0.25% RBCM, but not visible at 1–4% RBCM. However, no additional characteristics was observed. Instead, topographic information was very similar to surface roughness provided in Fig. S4. We have added the magnified images in the revised Supplementary Fig. 6.

Supplementary Fig. 6. Magnified AFM images of the AAO membrane with different RBCM concentrations. AFM, atomic force microscope; AAO, anodic aluminum oxide; RBCM, red blood cell membrane.

5. Pressure of 3 bar improved permeation of negatively charged BSA. Is the size of IgG being sole reason for enrichment of IgG or any other properties of IgG or interaction of IgG with RBCM could be plausible for enrichment of negatively charged IgG.

Answer)

To validate the size effect of IgG, we purchased Human IgG Fc fragments (401104, Sigma-Aldrich, USA) and prepared a concentration of 0.5 mg/mL with 1xPBS buffer. Since IgG is 150 kDa in size and the IgG Fc fragment is about 50 kDa, we observed no enrichment at a pressure of 3 bar. Through this experiment, we inferred that the effect of size of IgG is greater than that of other properties. We revised as *“To validate the size effect of IgG, we also tested Human IgG Fc fragments (50 kDa) and observed no enrichment of IgG Fc fragments at a pressure of 3 bar, inferring the size effect of negatively charged IgG on the enrichments.”* (line 193-196 and *See supplementary Fig. 9*)

Supplementary Fig. 9. Size effect of IgG on enrichment. (a) Enrichment process with BEETLES² (b) Enrichment zone of BEETLES² (c) No enrichment from IgG fragment (IgG Fc fragment molecular weight: 50 kDa). IgG: immunoglobulin G; BEETLES²: bioengineered enrichment tools for LFA with enhanced sensitivity and specificity.

We want to thank you again for your valuable comments, and we found the comments very helpful in clarifying the originality of our work. We hope the manuscript, with careful revisions, meets your high standards.

REVIEWER COMMENTS

Reviewer #1 (Remarks to the Author):

Dear Authors,

I hereby confirm that the manuscript has been improved after taking into account the proposals for modification and/or clarification.

I believe the manuscript meets now the scientific standards required for publication in Nature Communications.

Best regards

Olivier Vandenberg

Reviewer #2 (Remarks to the Author):

As stated previously, the manuscript with red blood cell membrane presents exciting ideas. However, the enrichment, tenability, selectivity, and specificity observations should be explained more scientifically and analytically. Why is the red blood cell membrane based on an AAO filter better than other preconcentration methods? After proper revision, the manuscript can be considered.

The authors have yet to respond to major comment #1

#1. The manuscript and the reported device are well characterized. I can predict that the reported device shows great potential for practical application for home diagnostics. However, a similar mechanism has been reported (Chen et al. Quantitative and ultrasensitive in situ immunoassay technology for SARS-CoV-2 detection in saliva. *Sci. Adv.* 8, eabn3481 (2022).) although the addition of red blood cell membranes can help to enrich the specific targets. Please show a competitive analysis concerning many similar works and justify why it has authentic merits.

While the reviewer appreciates the application of red blood cell membranes, critical analyses still need to be included.

Please show the competitive analysis.

1. "Competitive (or Comparative) Analysis" concerning other similar works (at least five different prior arts) with quantitative values is missing. Authors should compare with significant merits, such as detection enhancement, speed (total assay time including all operation steps, not just detection time), sensitivity (copies/ml of virus), selectivity, manufacturing time, and cost of the red blood cell membrane, reliability, reproducibility, and stability, etc. This paper does not include real-operation steps and time to recollect the filtered samples after enrichment and transfer to LFA.

2. Justify why it has authentic merits while other methods can enrich viruses by magnetic particles, selective acoustic separation, acoustic trapping, electrophoresis, dielectrophoresis, optical enrichment, etc.

3. Regarding the operation of the device, explain how bioengineered enrichment tools for lateral flow assays (LFAs) with enhanced sensitivity and specificity (BEETLES2) is better than other automated devices with better sensitivity, specificity, and speed. Please comment reliability of operation if BEETLES2 requires extra steps: (1) placing the membrane into the holder, (2) the viral sample injection with extra pressuring step, (3) viral sample enrichment, (4) reverse membrane processing for recollection of enriched sample, and (5) injection to commercial LFA.

4. Explain the challenging issues of BEETLES2 to integrate automated steps 1-4 so that the user can only drop the sample on the advanced LFA.

Reviewer #3 (Remarks to the Author):

The authors have been hyper-vigilant in addressing this reviewers concerns. This reviewer believe this has risen to the standards set by Nature Communications. ACCEPT.

Reviewer #4 (Remarks to the Author):

The authors have answered all the queries in detail. I would like to accept this manuscript in the current form.

Response letter

Reviewer #2

As stated previously, the manuscript with red blood cell membrane presents exciting ideas. However, the enrichment, tenability, selectivity, and specificity observations should be explained more scientifically and analytically. Why is the red blood cell membrane based on an AAO filter better than other preconcentration methods? After proper revision, the manuscript can be considered.

Answer) Thank you for your feedback on our manuscript. We have revised the manuscript based on your feedback and hope that the revised version will meet your expectations.

The authors have yet to respond to major comment #1

#1. The manuscript and the reported device are well characterized. I can predict that the reported device shows great potential for practical application for home diagnostics. However, a similar mechanism has been reported (Chen et al. Quantitative and ultrasensitive in situ immunoassay technology for SARS-CoV-2 detection in saliva. *Sci. Adv.* 8, eabn3481 (2022).) although the addition of red blood cell membranes can help to enrich the specific targets. Please show a competitive analysis concerning many similar works and justify why it has authentic merits.

While the reviewer appreciates the application of red blood cell membranes, critical analyses still need to be included.

Please show the competitive analysis.

1. “Competitive (or Comparative) Analysis” concerning other similar works (at least five different prior arts) with quantitative values is missing. Authors should compare with significant merits, such as detection enhancement, speed (total assay time including all operation steps, not just detection time), sensitivity (copies/ml of virus), selectivity, manufacturing time, and cost of the red blood cell membrane, reliability, reproducibility, and stability, etc. This paper does not include real-operation steps and time to recollect the filtered samples after enrichment and transfer to LFA.

2. Justify why it has authentic merits while other methods can enrich viruses by magnetic particles, selective acoustic separation, acoustic trapping, electrophoresis, dielectrophoresis, optical enrichment, etc.

Answer) To meet comments #1 and #2, we have revised the manuscript (*See lines 337-340*) with new *Supplementary Table 2*.

In *Supplementary Table 2*, we showed the summary of sample isolation and enrichment techniques with their properties, advantages, limitations, target virus, figures of merits (FOM), and analysis time for virus applications. We have added sentences such as "*Compared with other existing virus isolation and enrichment techniques, our BEETLES² has advantages in the permselectivity, tunability, powerful enrichment capability for both intact viruses and their N proteins, and applicability as a hand-held device (see **Table. S2** for the comparison of different virus isolation and enrichment techniques).*" (lines 337–340).

Supplementary Table 2. Summary of sample isolation and enrichment techniques with their properties, advantages, limitations, target virus, figures of merits (FOM), and analysis time.

Isolation and enrichment technique	Properties	Advantages	Limitations	Target virus	Figures of merits (FOM)	Analysis time (min)	Reference
Microbead	Charge	 - Short operating time - Not demanding external instruments 	 - Low throughput - Demanding operating steps 	SARS-CoV-2	Recovery rate: 33.9±13.8%	Sample prep. 30 min	1
	Immuno-affinity			Influenza A	Recovery rate: 90%	Sample prep. 35 min	2
	Immuno-affinity			Influenza A	Recovery rate: 50%	Sample prep. 25 min	3
Centrifugation	Density	 - High throughput - Simple operation steps 	 - Demanding external instruments - Poor isolation resolution (unable to discern similar density particles) 	SARS-CoV-2	Recovery rate: 69%	Sample prep. 1280 min	4
				murine hepatitis virus (MHV)	Recovery rate: 33.5±12.1%	Sample prep. 135 min	5
Dielectrophoresis	Size and polarizability	 - Low cost for device fabrication - Short operating time - Automation 	 - Demanding volume dependent force - Demanding external instruments - Buffer dependent performance (reduced performance in high ionic strength buffer) 	T7 phage	Sensitivity: 10 ⁴ particles/mL	Sample prep. +Analysis 5 min	6
				MS2 virus	ANOVA followed Tukey post hoc test: p-value < 0.035	Sample prep. +Analysis 1 min	7

Acoustofluidics	Size and acoustic contrast factor	- Excellent biocompatibility	- Demanding volume dependent force - Demanding external instruments	Japanese encephalitis virus	Separation efficiency: 99%	NA	8
		- Precise particle manipulation (trapping, enrichment, isolation) - Buffer independent performance - Automation		Dengue virus	Extract efficiency: 90%	Sample prep. 9 min	9
Electrokinetics	Size and charge	- Simple operation - High throughput	- Buffer dependent performance (reduced performance in high ionic strength buffer)	Baculovirus (AcNPV)	Concentration rate: 1.2	NA	10
Membrane filtration	Size	- Hand actuated - Short operating time - Commercially available	- Clogging - Poor isolation resolution (unable to discern similar size particles)	Influenza A	virus capture efficiency: 96.5±0.5%	Sample prep. 1 hour	11
				Hepatitis C virus	Enrichment efficiency: 91%	NA	12
		- Automation - Short operating time	- Demanding external instruments - Enrichment capability only for intact viruses - Limited in commercial inactivated lysate buffers (i.e., VTM)	SARS-CoV-2	Preconcentration: 40-folds No Ct value increase for lysed virus	Sample prep. 3 min + Analysis <17 min	13
	Size and charge	- Charge-based isolation and enrichments (permselectivity) - Size-based isolation and enrichments - Powerful enrichment capability for both intact viruses and their N proteins - Short operating time - Hand actuated	- A leak-free design is needed. - Volume issue in serological rapid kit.	SARS-CoV-2 (Delta, omicron) & Influenza A/B	Preconcentration: up to 34.3-folds Ct value increase up to 5.1	Sample prep. 3 min + Analysis 15 min	This works

Reference

1. Lázaro-Perona F, *et al.* Evaluation of two automated low-cost RNA extraction protocols for SARS-CoV-2 detection. *PLoS One* **16**, e0246302 (2021).
2. Bai Z, *et al.* Rapid Enrichment and Ultrasensitive Detection of Influenza A Virus in Human Specimen using Magnetic Quantum Dot Nanobeads Based Test Strips. *Sensors Actuators B: Chem* **325**, 128780 (2020).
3. Li G, *et al.* Influenza Virus Precision Diagnosis and Continuous Purification Enabled by Neuraminidase-Resistant Glycopolymers-Coated Microbeads. *ACS Applied Materials & Interfaces* **13**, 46260-46269 (2021).
4. Gias E, Nielsen SU, Morgan LAF, Toms GL. Purification of human respiratory syncytial virus by ultracentrifugation in iodixanol density gradient. *J Virol Methods* **147**, 328-332 (2008).
5. Ahmed W, *et al.* Comparison of virus concentration methods for the RT-qPCR-based recovery of murine hepatitis virus, a surrogate for SARS-CoV-2 from untreated wastewater. *Science of The Total Environment* **739**, 139960 (2020).
6. Yeo W-H, Lee H-B, Kim J-H, Lee K-H, Chung J-H. Nanotip analysis for dielectrophoretic concentration of nanosized viral particles. *Nanotechnology* **24**, 185502 (2013).
7. Han C-H, Woo SY, Bhardwaj J, Sharma A, Jang J. Rapid and selective concentration of bacteria, viruses, and proteins using alternating current signal superimposition on two coplanar electrodes. *Sci Rep* **8**, 14942 (2018).
8. Liu Z, *et al.* Fluorescent labeling based acoustofluidic screening of Japanese encephalitis virus. *Sensors Actuators B: Chem* **322**, 128649 (2020).
9. Fong EJ, *et al.* Acoustic focusing with engineered node locations for high-performance microfluidic particle separation. *Analyst* **139**, 1192-1200 (2014).
10. Mogi K, Hayashida KEI, Honda A, Yamamoto T. Development of Virus Concentration Device by Controlling Ion Depletion Zone for Ultrasensitive Virus Sensing. *Electronics and Communications in Japan* **100**, 56-63 (2017).
11. Yeh Y-T, *et al.* Tunable and label-free virus enrichment for ultrasensitive virus detection using carbon nanotube arrays. *Science Advances* **2**, e1601026 (2016).
12. Jeon G, Jee M, Yang SY, Lee B-y, Jang SK, Kim JK. Hierarchically Self-Organized Monolithic Nanoporous Membrane for Excellent Virus Enrichment. *ACS Applied Materials & Interfaces* **6**, 1200-1206 (2014).
13. Chen Y, Liu F, Lee LP. Quantitative and ultrasensitive in situ immunoassay technology for SARS-CoV-2 detection in saliva. *Science Advances* **8**, eabn3481 (2022).

3. Regarding the operation of the device, explain how bioengineered enrichment tools for lateral flow assays (LFAs) with enhanced sensitivity and specificity (BEETLES²) is better than other automated devices with better sensitivity, specificity, and speed. Please comment reliability of operation if BEETLES² requires extra steps: (1) placing the membrane into the holder, (2) the viral sample injection with extra pressuring step, (3) viral sample enrichment, (4) reverse membrane processing for recollection of enriched sample, and (5) injection to commercial LFA.

Answer) Regarding the reliability of operation, while the current design works properly, we do not think the suggested design of hand-powered portable syringes is not the final version of commercializing items. To show the commercializing issues, we have added a paragraph at the end of the Results at the first-round revision according to two reviewer's suggestion as *"To commercialize BEETLES², we should address the following issues: 1) pressure endurable design. Since the BEETLES² assay is performed under applied pressure, no liquid sample leakage can be tolerated. Especially when handling infectious samples, a leak-free design is needed for safety. 2) The volume issue in serological rapid diagnostic tests. For serological rapid diagnostic tests, 1–2 drops (<100 µL) a typically acquired from a finger prick; therefore, we need to develop devices for blood-based tests with small sample volumes. A microfluidic device integrated with the BEETLES² membrane is an ideal candidate for serological assays, 3) Shelf life is a key parameter to commercialize BEETLES². We obtained stability, for up to 15 days, under room temperature. However, the expected shelf life can be extended further with additional tests (Fig. S13)."* (line 340–350)

4. Explain the challenging issues of BEETLES² to integrate automated steps 1-4 so that the user can only drop the sample on the advanced LFA.

Answer) In previous answers for comment #3, we showed the challenging issues. Since affordability (cheaper LFA) is the key parameter in the immunoassay-based rapid test, we think the manually operated hand-powered BEETLES² could be the better solution than the automated systems. Therefore, we designed "hand-powered portable syringes" as a low-cost alternative for point-of-care diagnostics.

However, we need to develop BEETLES² with an automated system for molecular diagnostics. As the reviewer knows, sample preparation in molecular diagnostics is a complex and critical step. Despite recent advancements, several limitations still need to be addressed, including

complexity, standardization, inhibitors, and limited specificity. We are now trying to design BEETLES² with automated steps for molecular diagnostics, including exosome and miRNA-based diagnostics.

We want to thank you again for your valuable comments. We already cleared all the other reviewer's comments, so the answer to comment #4 could be the last one from the four reviewer's comments. We found your comments very helpful in clarifying the originality of our work. We hope the manuscript, with careful revisions, meets your high standards.